# Ad-Hoc Human-AI Coordination Challenge

**Tin Dizdarević** [1]  **Ravi Hammond** [1]  **Tobias Gessler** [1]  **Anisoara Calinescu** [2]  **Jonathan Cook** [1]  **Matteo Gallici** [3]
**Andrei Lupu** [1]  **Jakob Nicolaus Foerster** [1]

## Abstract

Achieving seamless coordination between AI agents and humans is crucial for real-world applications, yet it remains a significant open challenge. Hanabi is a cooperative card game featuring imperfect information, constrained communication, theory of mind requirements, and coordinated action – making it an ideal testbed for human-AI coordination. However, its use for human-AI interaction has been limited by the challenges of human evaluation. In this work, we introduce the Ad-Hoc Human-AI Coordination Challenge (AH2AC2) to overcome the constraints of costly and difficult-to-reproduce human evaluations. We develop *human proxy agents* on a large-scale human dataset that serve as robust, cheap, and reproducible human-like evaluation partners in AH2AC2. To encourage the development of data-efficient methods, we open-source a dataset of 3,079 games, deliberately limiting the amount of available human gameplay data. We present baseline results for both two- and three- player Hanabi scenarios. To ensure fair evaluation, we host the proxy agents through a controlled evaluation system rather than releasing them publicly. The code is available at https://github.com/FLAIROx/ah2ac2.

## 1. Introduction

Human–AI interaction is rapidly advancing due to significant AI progress and its growing integration into daily life (Rawas, 2024). Effective coordination between humans and AI in complex settings becomes crucial as AI agents become more sophisticated, capable, and prevalent. This interaction spans a wide range of domains, from collaborative decision-making in healthcare (Asan et al., 2020) to shared control in autonomous vehicles (Bansal et al., 2018) and robotics (Haarnoja et al., 2018; Ahn et al., 2024) as well as advanced digital assistants (Gemini Team, 2024; Bai et al., 2022). The ultimate goal is to create AI agents that are not limited to solving problems independently but can also work effectively with humans to complete tasks in human-compatible ways (Russell, 2019; Carroll et al., 2019).

Traditional approaches to training AI agents in simulation frequently employ *self-play* (SP) where agents train under a joint policy that controls the strategies of all players (Samuel, 1959; Tesauro, 1994). While SP has been successful in competitive games like chess and Go (Silver et al., 2016), in cooperative settings this approach can lead to agents that overfit to specific strategies, limiting their ability to generalise to novel partners (Carroll et al., 2019). In human-AI coordination scenarios, forming rigid conventions is particularly problematic, as doing so can pose safety risks where humans are unable to adapt appropriately (Bard et al., 2019).

Despite the growing importance of human-AI coordination, the field lacks standardised benchmarks that accurately reflect the complexities of interacting with humans in complex, partially observable settings. Existing evaluation methods often rely on closed datasets and proprietary proxy agents (Bakhtin et al., 2022; Jacob et al., 2022), which hinder reproducibility and broad-based progress. Without accessible and robust benchmarks, it is challenging to measure advancements consistently. Furthermore, the scarcity of open-source human-AI coordination datasets limits the ability of researchers to develop and test innovative, data-efficient algorithms that are essential for real-world applications where large-scale human data may not be readily available.

Hanabi is an established, fully cooperative benchmark environment that involves imperfect information, limited communication, theory of mind, and the necessity for coordination among different players to achieve a shared goal (Bard et al., 2019). These characteristics make Hanabi a solid testbed for evaluating human-AI coordination. While previous research in Hanabi has used held-out sets of human data and human proxy agents to evaluate human-AI coordination (Hu et al., 2022; 2021; Lupu et al., 2021), these datasets and proxy agents have thus far remained closed-source and un-

---

[1]FLAIR, University of Oxford, Oxford, UK [2]Department of Computer Science, University of Oxford, UK [3]Universitat Politècnica de Catalunya, Barcelona, Spain. Correspondence to: Tin Dizdarević <tin@robots.ox.ac.uk>.

*Proceedings of the $42^{nd}$ International Conference on Machine Learning*, Vancouver, Canada. PMLR 267, 2025. Copyright 2025 by the author(s).

available to the wider research community. To address these problems, we introduce the *Ad-Hoc Human-AI Coordination Challenge* (AH2AC2) as a standardised way to evaluate human-AI coordination in Hanabi. Specifically, we develop human proxy agents through a combination of behavioural cloning (BC) and regularised reinforcement learning (RL). We first train the BC component on a large-scale dataset of human gameplay, comprising 101,096 two-player games and 46,525 three-player games from the hanab.live community. We then refine the BC policy using Independent Proximal Policy Optimisation (IPPO) (de Witt et al., 2020; Schulman et al., 2017) with a regularisation term that encourages adherence to human-style play (Bakhtin et al., 2022; Hu et al., 2022; Cornelisse & Vinitsky, 2024). Our empirical evaluation shows that these human proxy agents outperform pure imitation learning while maintaining human-like behaviour. These human proxy agents serve as robust, cheap and reproducible evaluation partners in AH2AC2.

To ensure AH2AC2 evaluation integrity and to prevent over-fitting by the community, we withhold public access to the human proxy agents and their large-scale training dataset, and only open-source a limited dataset for both two- and three-player settings. This also encourages research on methods that are data-efficient with respect to human data. We also provide various baselines: first, zero-shot coordination methods such as Off-Belief Learning (OBL) (Hu et al., 2021), which operate without human data; second, data-dependent approaches such as best response to behavioural cloning policy (BR-BC) (Carroll et al., 2019); third, the first evaluation of Fictitious Co-Play (FCP) (Strouse et al., 2021) in Hanabi, which serves as a baseline for population-based methods; fourth, we develop a DeepSeek-R1 (DeepSeek-AI et al., 2025) Hanabi agent, to provide insights into the current capabilities of Large Language Models (LLMs) for human-AI coordination.

In summary, our key contributions are:

- The first open-source Hanabi human gameplay dataset, containing 1,858 two-player and 1,221 three-player games.
- High-performing human proxy agents for both two-player and three-player Hanabi settings, using BC on a closed-source large-scale human play dataset (over 100k games) combined with regularised RL.
- Our evaluation protocol, where we host proxy agents behind an API, paired with a public leaderboard to track research progress. API access requires the pre-registration of experiments, a gold standard for empirical research.
- A comprehensive set of baselines for both two- and three-player settings, spanning zero-shot coordination, data-dependent, and population-based methods, alongside the DeepSeek-R1 Hanabi agent, which specifically provides an assessment of LLM performance in human-AI coor-

dination; these baselines collectively highlight the significant difficulty current methods face in building human-compatible agents for partially observable environments.

## 2. Background

**Dec-POMDP** We consider a decentralised partially observable Markov decision process (Dec-POMDP) (Oliehoek, 2012), defined as a 9-tuple $(\mathcal{S}, n, \{\mathcal{A}^i\}_{i=1}^n, \{\mathcal{O}^i\}_{i=1}^n, \mathcal{T}, \mathcal{R}, \{\mathcal{U}^i\}_{i=1}^n, H, \gamma)$. $\mathcal{S}$ is the finite state space and $n$ is the number of agents. $\mathcal{A}^i$ and $\mathcal{O}^i$ are the local action and observation spaces for agent $i$, and $\mathcal{A} := \times_{j=1}^n \mathcal{A}^i$, $\mathcal{O} := \times_{i=1}^n \mathcal{O}^i$ are the joint action and observation spaces. The transition function $\mathcal{T} : \mathcal{S} \times \mathcal{A} \times \mathcal{S} \to [0, 1]$ defines the probability of transitioning to state $s_{t+1}$ when taking joint action $a_t = (a_t^1, ..., a_t^n)$ in state $s_t$. The agents receive the reward $r_{t+1} = \mathcal{R}(s_{t+1}, a_t)$, and agent $i$ receives the local observation $o_{t+1}^i$ with probability $\mathcal{U}^i(s_{t+1}, a_t, o_{t+1}^i)$. $\gamma \in [0, 1]$ is the discount factor, and $H$ is the horizon. i.e. $s_H$ is always a terminal state.

The local action-observation history (AOH) of player $i$ is defined as $\tau_t^i = (a_0^i, o_1^i \ldots, o_{t-1}^i, a_{t-1}^i, o_t^i)$, and the joint AOH is defined as $\tau = (\tau^1, ..., \tau^n)$. Each player $i$ selects a local action $a_t^i$ according to a local policy $\pi^i(a_t^i|\tau_t^i)$. The joint policy $\pi = (\pi^1, ..., \pi^n)$ then selects joint action $a_t$ with probability $\pi(a_t|\tau_t) = \prod_{j=1}^n \pi^i(a_t^i|\tau_t^i)$. Given a joint policy $\pi$, the expected return is defined as $J(\pi) = \mathbb{E}_\pi \left[ \sum_{t=0}^{H-1} \gamma^t r_{t+1} \right]$.

**Zero-shot Coordination and Ad-Hoc Teamplay** In many real-world scenarios, AI agents must coordinate with unseen partners, including humans. Traditional cooperative multi-agent RL uses SP training, where agents train together to maximize the expected return (Bard et al., 2019; Carroll et al., 2019). However, SP training often leads to specialized communication protocols that fail with independently trained agents, including humans. Thus, training in SP is not a solution to the challenges of human-AI coordination (Carroll et al., 2019; Strouse et al., 2021). Zero-shot coordination (ZSC) addresses this by training agents to collaborate effectively with new partners which were trained with the same algorithm (Hu et al., 2020; Treutlein et al., 2021).

Ad-hoc teamplay assesses an agent's ability to cooperate with unfamiliar teammates at test time (Stone et al., 2010). Like ZSC, agents are evaluated with partners they have not trained with, but unlike ZSC, teammates may use different training algorithms. In this work, we focus on ad-hoc teamplay involving human and human-like agents.

**Hanabi** Hanabi is a cooperative card game where players can see the cards in each other's hands, but not in their own,

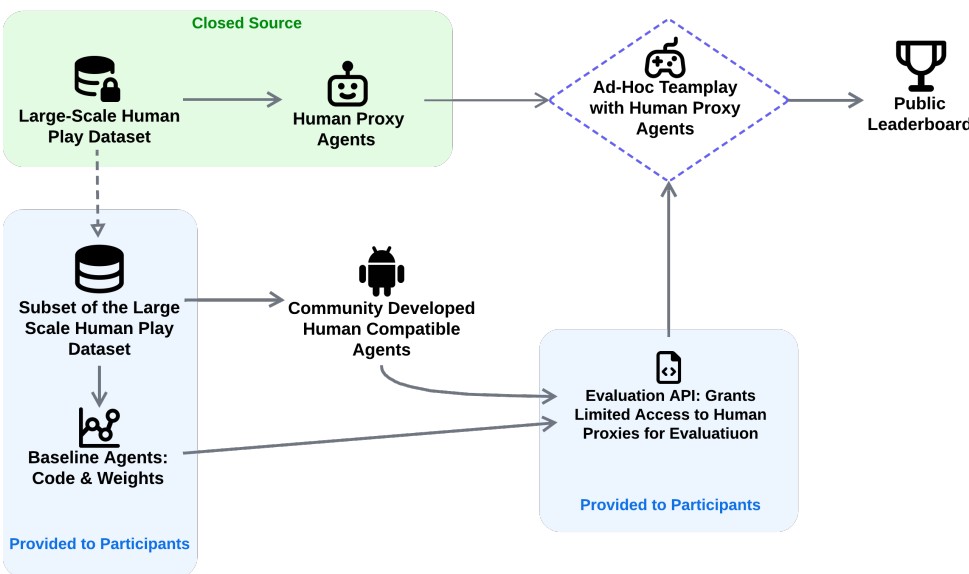

*Figure 1.* Ad-Hoc Human-AI Coordination Challenge (AH2AC2)

and thus rely on others to give interpretable hints on which cards to play or discard. It is designed for 2-5 players, but we restrict ourselves to two- and three- player settings in this work due to the availability of human gameplay data for these configurations. The maximum score in the game is 25, but it cannot be achieved for every shuffling of the deck. If the team of players makes three mistakes in total, the game ends with a score of 0. For more details on Hanabi we refer to Appendix A.1.

## 3. Related Work

Hanabi was introduced as a benchmark encompassing both SP and ad-hoc teamplay (Bard et al., 2019), but significant progress has been primarily confined to SP, with methods like SPARTA (Lerer et al., 2019) achieving near-perfect 24.61/25 points on average. However, agents trained in SP often rely on specialised conventions, leading to poor generalisation when paired with novel teammates. Consequently, ad-hoc team play, especially with humans, presents a more demanding and unsolved challenge.

A central issue when evaluating agent abilities for ad-hoc coordination is the selection of test-time partners. One approach to tackle this issue is to ensure the diversity of policies, thereby minimising the possibility of favouring any specific agent. For example, Cui et al. (2023) introduce ADVERSITY that aims to produce highly skilled and reasonable policies that play according to diverse conventions. Another approach is to focus on a set of policies that hold intrinsic value, with the most natural choice being human policies. Coordinating with humans can be seen as a

specialised form of ad-hoc teamplay, where the set of test policies comprises human strategies, which are inherently valuable due to their real-world relevance.

Ad-hoc human-AI coordination in Hanabi is explored in many previous works, with each presenting a different methodology for acquiring human proxy agents and evaluating human-AI coordination capabilities (Hu et al., 2021; 2020; Cui et al., 2023). However, there is no standard approach for ad-hoc human-AI coordination evaluation in the existing literature. Therefore, our work addresses this gap and proposes the AH2AC2.

Recent works have empirically shown that augmenting imitation learning methods with regularised RL creates stronger and more reliable policies. Multiple works have shown that regularised RL leads to policies that are more compatible with existing social conventions of the human reference group (Jacob et al., 2022; Bakhtin et al., 2022; Hu et al., 2022). Recently, (Cornelisse & Vinitsky, 2024) extended these works to the driving setting, where the authors show that data-driven regularisation leads to human-compatible policies. Our work builds upon these findings to build strong and human-like human proxy agents that enable AH2AC2.

Nekoei et al. (2023) urge the multi-agent RL community to address few-shot adaptation alongside zero-shot coordination (ZSC). They show that current ZSC algorithms struggle to adapt to new partners. To connect AH2AC2 with few-shot coordination, we introduce data-limited settings. Unlike traditional few-shot scenarios, our approach involves offline, one-sided adaptation: human proxy agents maintain fixed behaviour during testing, and we use a small sample of

human play data for training. This emphasizes the challenge of coordinating with human-like agents, where only the AI agent adapts due to the constraints of human behaviour (Stone et al., 2010).

# 4. Ad-Hoc Human-AI Coordination Challenge (AH2AC2)

This section outlines our proposed challenge, which consists of two key evaluation regimes: (a) evaluation with a set of human proxy agents. This requires participants to develop human-compatible agents based on a small provided dataset of human gameplay, and (b) a human action prediction task on an unseen, closed-source set of games.

## 4.1. Methodology Overview

We collected a comprehensive dataset from the hanab.live platform, comprising of 101,096 two-player games and 46,525 three-player games. All games adhere to H-group conventions, a set of hand-crafted strategies used by Hanabi players on hanab.live. It is important to note that H-group Conventions are not a single strategy. Instead, it is helpful to think of H-Group Conventions as a collection of different strategies and techniques that players learn and combine. Players often mix and adapt these strategies within a single game, depending on the players' strength. Therefore, the dataset itself naturally contains a variety of playstyles. Therefore, the use of H-Group Conventions as a foundation does not overly constrain the strategic variety of our human proxies. Details about data composition and splits are provided in Appendices A.5 and A.4.

As a part of the AH2AC2, we open source 3,079 games from the large-scale dataset — 1,858 two-player and 1,221 three-player games. Participants are allowed to use these open-sourced games when tackling the AH2AC2. Key statistics of the open-sourced dataset are summarised in Table 1.

Using the entire dataset we develop human-proxy agents that act as standard and cheap test partners for ad-hoc human-AI coordination evaluation. To prevent overfitting, we host the human proxies behind an API instead of releasing them publicly. Participants have to pre-register an evaluation which gives them access to to 1,000 evaluation games with our human proxies. This controlled access ensures consistency across submission and pre-registration of experiments is the gold standard for empirical science. The candidate agent's performance is evaluated based on the mean and median scores achieved across 1,000 games with human proxies and will be published on our leaderboard.

In the second (optional) part of the challenge, we also assess the agent's ability to predict human actions in an unseen set of human-played games. We evaluate performance using the teacher-forced cross-entropy loss.

## 4.2. Evaluation Protocol

### PART 1 OF AH2AC2: COORDINATION WITH HUMAN PROXIES

We develop four human proxy agents for evaluating ad-hoc human-AI coordination: two for two-player Hanabi and two for three-player. These agents are trained using Human-Data-Regularised IPPO (HDR-IPPO), a procedure combining BC and regularised IPPO (de Witt et al., 2020). First, BC policies are trained on a large-scale dataset of human gameplay - 101,096 two-player games and 46,525 three-player games. Because BC alone struggles to generalise to unseen game states (Carroll et al., 2019; Hu et al., 2022; Bakhtin et al., 2022; Cornelisse & Vinitsky, 2024), we then refine them through regularised SP using IPPO. The regularisation ensures the final policies remain close to human play styles. We provide further details regarding PPO and IPPO in Appendix A.2.

When training human proxy agents using HDR-IPPO, we learn a parameterised local policy, denoted as $\pi_\theta^{HP}$. As a first step, we train a local BC policy $\pi_\theta^{BC}$, which, given Hanabi's discrete action space, translates into a classification task: the features are local AOHs $\tau_t^i$, and the labels are ground truth local actions $a_{t+1}^i$ given.

To capture the sequential nature of the actions and observations, we model $\pi_\theta^{BC}$ through an LSTM-based architecture (Hochreiter & Schmidhuber, 1997). Critically, fixed neural parameters do not imply static behaviour; our proxies condition on the history of the game, which includes partner actions. When the proxy agents see an unexpected action or observation, from that action-observation history onward, they will account for the fact that the other agent is using a different convention. We train the BC model through supervised learning, minimising the standard cross-entropy loss between the predicted action distribution and the ground truth human actions.

At the end of each training epoch, we compute the average SP score of $\pi_\theta^{BC}$ over 5000 games and store the parameters $\theta'$, that yield the highest average SP score. During each of those SP evaluations, each agent, at every timestep, selects the local action with the highest predicted probability according to the local policy, i.e. $a_t^i = \arg\max_a \pi_\theta^{BC}(a^i|\tau_t^i)$.

In the second step of the HDR-IPPO method we leverage the baseline BC policy, $\pi_{\theta'}^{BC}$, to guide the training of a more robust policy, $\pi_\theta^{HP}$. First, we initialize the weights of $\pi_\theta^{HP}$ to $\theta'$. Next, to encourage the final policy to remain close to the human-like behaviour exhibited by $\pi_{\theta'}^{BC}$, we introduce the KL (Kullback & Leibler, 1951) regularisation term

$$D_{\text{KL}}(\pi_{\theta'}^{BC}(\cdot|\tau_t^i)||\pi_\theta^{HP}(\cdot|\tau_t^i)). \tag{1}$$

We build upon previous works (Cornelisse & Vinitsky, 2024; Jacob et al., 2022; Hu et al., 2022; Bakhtin et al., 2022),

*Table 1.* Statistics for open-sourced data. The table presents the minimum, maximum, average, median, and standard deviation for both game scores and game lengths in each setting.

| Setting | Metric | Min | Max | Avg | Median | Std |
|---|---|---|---|---|---|---|
| 1,858 Two-Player Open-Sourced Games | Scores | 13 | 25 | 23.37 | 24 | 1.86 |
| | Game Lengths | 52 | 76 | 65.45 | 66 | 3.35 |
| 1,221 Three-Player Open-Sourced Games | Scores | 14 | 25 | 23.25 | 24 | 1.91 |
| | Game Lengths | 45 | 67 | 57.86 | 58 | 3.38 |

which demonstrate that introducing KL regularisation in this manner preserves human-likeness of the BC policy while being more robust.

This KL term is then added to the standard IPPO (de Witt et al., 2020) objective $\mathcal{L}_t^{\text{IPPO}}(\theta)$, with a regularisation weight $\lambda \in [0, 1]$:

$$\mathcal{L}_t^{\text{HDR-IPPO}}(\theta) = (1 - \lambda) \cdot \mathcal{L}_t^{\text{IPPO}}(\theta)$$
$$+ \lambda \cdot D_{\text{KL}}\left(\pi_{\theta'}^{BC}(\cdot|\tau_t^i)||\pi_\theta^{HP}(\cdot|\tau_t^i)\right). \quad (2)$$

Further details and a complete list of hyperparameters used for training human proxy agents are available in Appendix A.5.

While our approach builds on prior research (Cornelisse & Vinitsky, 2024; Jacob et al., 2022; Bakhtin et al., 2022; Hu et al., 2022), we design the following experiments to empirically validate that the generated human proxy agents exhibit human-like behaviour in Hanabi:

- **Cross-Play between Human Proxies and BC Policies:** We test how well the human proxy agents coordinate with baseline BC policies, which closely follow human conventions but may lack generalisation. Strong performance in this cross-play setting confirms that our HDR-IPPO agents maintain human-compatible conventions while improving upon the generalisation limitations of pure BC.
- **Action Prediction Performance of Human Proxies**: We compare the performance of the BC policies on the test datasets before HDR-IPPO training procedure with that of the final human proxies at the end of HDR-IPPO procedure. This comparison allows us to evaluate whether the agent retains human-like conventions present in the dataset.
- **Behaviour Analysis of Human Proxies**: We use the behavioural metrics *Information per Play* (IPP) and *Communicativeness*, as proposed by (Canaan et al., 2020), to evaluate both the trajectories in the dataset and those generated in SP by the human proxies. These metrics are straightforward to measure and are recognized as strategically important. By making this comparison, we demonstrate that human proxies display behaviours consistent with those in the human play dataset.

Our human proxies serve as standardised, robust and cheap partners for human-AI coordination evaluation. In the context of AH2AC2, human proxies play a pivotal role in benchmarking AI performance against human-like behaviour. This setup not only facilitates the assessment of AI adaptability and robustness but also ensures that evaluations are scalable and reproducible.

Additionally, we present an ablation study to examine the impact of the HDR-IPPO KL regularisation term in Appendix A.8. This analysis explores the effects of varying regularisation strength on the learned policies, offering insights into the role of this component. We defer this discussion to the appendix to maintain focus on the properties of the developed human proxy agents within the main text.

PART 2 OF AH2AC2: ACTION PREDICTION CHALLENGE

Beyond the primary human-AI ad-hoc coordination challenge, we introduce an action prediction task. Although human-compatible play does not strictly ensure accurate action prediction, successfully predicting human actions further demonstrates human-like behaviour. The action prediction dataset consists of a held-out portion of the human data used to train the human proxies.

We evaluate performance using the teacher-forced cross-entropy loss since we aim to quantify the difference between the predicted action distribution and the true human actions. Lower cross-entropy loss indicates better alignment with human decision-making. Participating agents receive a local action-observation history and must predict the action taken by the human player at each timestep.

### 4.3. Evaluation API and Leaderboard

To facilitate participation and ensure fair evaluation, we host the human proxy agents and provide access through a dedicated evaluation API.

We have established a dedicated website for the AH2AC2 challenge at https://ah2ac2.com/. To initiate the evaluation process, participants must fill out a form to register for access to the evaluation phase. Once registered, we provide participants with a private key, which grants restricted access

to our human proxies. This key allows a one-time evaluation run, strictly limited to 1,000 games. Once the evaluation with human proxies is finished, participants get limited access to the test dataset through the API we provide. Upon completion, results are published on the challenge leaderboard. Furthermore, our evaluation API allows interaction with the proxy agents while restricting access to global game state information, enforcing the partial observability inherent to Hanabi.

# 5. Analysing and Validating the Human-Proxies

This section is organised as follows. First, we evaluate the self-play performance of our human proxy agents, demonstrating significant improvements compared to the BC policies. Second, we validate the human-likeness of our human proxy (HP) agents. This is done through cross-play experiments with BC policies, by an evaluation of the action prediction performance of the HP policies on held-out datasets of human gameplay, and with an analysis of behavioural metrics.

## 5.1. Self-Play Scores of Human Proxy Agents

Table 2 presents the SP scores of our final human proxy agents and their improvement over the initial BC policies. The performance gains from regularised RL are particularly pronounced in the three-player setting, where the BC policies trained on limited data frequently lose all their lives, resulting in a large proportion of zero-score games and overall bad performance. For instance, in a three-player SP evaluation, BC agents scored zero in 70.92% of games. With the help of regularised RL, human proxies score zero points only on 0.27% of the games. This highlights the robustness achieved through regularised RL. Moreover, we observe a larger number of perfect-score games for all agents, compared to BC counterparts.

Furthermore, Figure 2a and 2b illustrate the cross-play results for two-player and three-player setting human proxy agents. We observe consistent scores across different pairings, suggesting that the agents have converged to compatible strategies despite variations in their architectures and regularisation strengths.

## 5.2. Validating Human-Likeness of Human Proxy Agents

**Cross-Play between Human Proxies and BC Policies.** Behavioural cloning (BC) policies are closely aligned with human conventions but lack generalization capabilities. However, we anticipate that when BC policies are paired with human proxies, the resulting cross-play scores will be substantially higher than the BC policies' SP scores. This

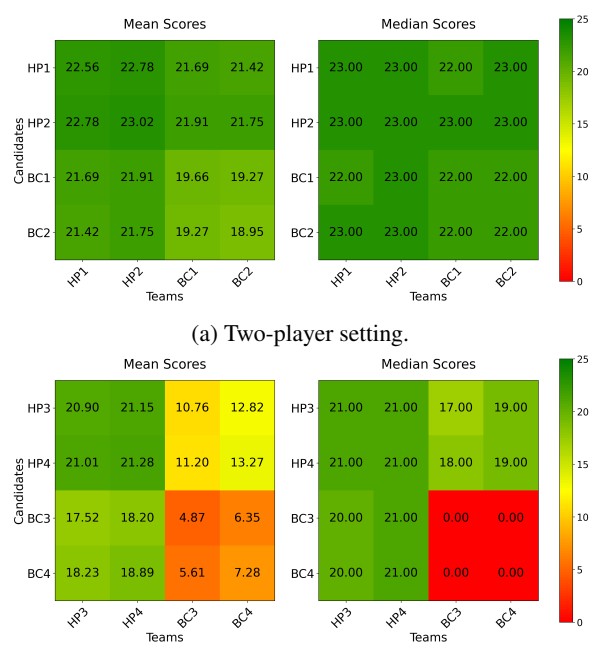

(a) Two-player setting.

(b) Three-player setting.

*Figure 2.* Cross-play performance matrices for human proxy agents and corresponding BC policies in two-player and three-player settings. Strong cross-play performance between two policies indicates that policies converged to compatible conventions. Each element $(i, j)$ represents the average score achieved by different team configurations, over all possible permutations of player positions. For the three-player setting team comprises two instances of agent $j$ and one instance of agent $i$. The average score for each team is calculated based on 15,000 games per permutation.

expectation is confirmed in Figures 2a and 2b. By comparing median and mean scores, we observe that agents either coordinate exceptionally well or fail completely. These results suggest that HDR-IPPO agents have not only learned to play effectively amongst each other but have also retained strategies employed by BC agents (i.e. agents trained solely on human data).

In the three-player setting, mean scores are significantly lower than median scores, especially when two BC policies team with one human proxy. This suggests many zero-score games and highlights the brittleness of BC policies in unfamiliar scenarios. Despite this, median scores remain high despite BC policies' poor single-player performance. When two BC policies are paired with a stronger human proxy, median scores increase substantially, reaching 17 or higher in all configurations.

These cross-play experiments provide evidence that human proxy agents have successfully learned to play the game at a high level while maintaining the ability to interact effectively with agents that utilize strategies derived purely from human demonstrations.

*Table 2.* SP evaluation results for the four human proxies. We compare human proxy results with the results of the respective BC policies. For SP evaluation, we report mean $\pm$ SE over 5,000 games.

| Metric | $\pi_{\theta^1}^{HP}$ | $\pi_{\theta^2}^{HP}$ | $\pi_{\theta^3}^{HP}$ | $\pi_{\theta^4}^{HP}$ |
|---|---|---|---|---|
| Mean Self-Play Score | $22.55 \pm 0.03$ | $22.97 \pm 0.03$ | $20.88 \pm 0.03$ | $21.21 \pm 0.03$ |
| Improvement over BC | 3.0 | 4.0 | 15.7 | 13.9 |
| Perfect Games (%) | 23.86% | 29.66% | 2.76% | 3.88% |
| Perfect Games BC (%) | 16.12% | 19.88% | 1.34% | 1.80% |
| Zero-Score Games (%) | 0.10% | 0.04% | 0.34% | 0.20% |
| Zero-Score Games BC (%) | 11.42% | 17.70% | 75.82% | 66.02% |

**Action Prediction Performance of Human Proxies.** Next, we evaluate the action prediction performance of human proxies using a test set that was excluded from the training of the BC agents. The results, reported in Table 3, demonstrate that the human proxies achieve similar accuracy and loss metrics on the test sets as the BC policies. In many Hanabi game scenarios, multiple human-like actions are possible. We thus report the Top-10% and Top-20% accuracies, which represent the probability that the ground-truth action is among the top 10%, 20% of most likely actions under the human proxy policies. This metric corresponds to top-2, top-4, and top-3, top-6 accuracies for two-player and three-players, respectively.

*Table 3.* Loss and accuracy of the four human proxy policies on the test dataset. Compared to respective BC policies, human proxies significantly improve in SP, while still fitting the test sets well. Additionally, the high Top-10% and Top-20% accuracies suggest that our proxies are indeed human-like.

| Metric | $\pi_{\theta^1}^{HP}$ | $\pi_{\theta^2}^{HP}$ | $\pi_{\theta^3}^{HP}$ | $\pi_{\theta^4}^{HP}$ |
|---|---|---|---|---|
| Number of Players | 2 | 2 | 3 | 3 |
| Accuracy | 0.63 | 0.63 | 0.43 | 0.44 |
| Accuracy Difference to BC | -0.03 | -0.08 | -0.08 | -0.07 |
| Loss | 0.53 | 0.54 | 0.63 | 0.60 |
| Loss Difference to BC | +0.05 | +0.07 | +0.08 | +0.06 |
| Top-10% Accuracy | 0.82 | 0.82 | 0.71 | 0.73 |
| Top-20% Accuracy | 0.95 | 0.95 | 0.87 | 0.88 |

**Behaviour Analysis of Human Proxies.** We assess the behaviour against a large-scale human dataset using two metrics from (Canaan et al., 2020): **IPP** and **Communicativeness**. **IPP** (Information per Played Card) measures the information an agent has about each card it plays: for every card played, we track which attributes (colour and/or rank) are known (0, 1, or 2), average these values across all played cards, and normalize by dividing by 2 to obtain a score between 0 and 1. **Communicativeness** measures the proportion of turns where an agent gives a hint when a hint token is available, quantifying how often an agent chooses to communicate. These metrics were computed from 50,000 human-proxy trajectories in SP and the exten-

sive human play dataset. As shown in Table 4, both metrics are nearly identical across human proxies and the human dataset, indicating similar behaviour and strategy.

*Table 4.* Comparison of behaviour features between the large-scale dataset and 50,000 trajectories generated by human proxies. **IPP** (Information Per Played card) measures the average number of known attributes (colour and/or rank) per played card, normalized between 0 and 1. **Communicativeness** quantifies the proportion of turns in which an agent chooses to give a hint when a hint token is available. The metrics are nearly identical across data sources, indicating similar behaviour and strategy.

| Setting | Source | IPP | Communicativeness |
|---|---|---|---|
| 2P | 2P Dataset | 0.44 | 0.47 |
| | $\pi_{\theta^1}^{HP}$ | 0.43 | 0.45 |
| | $\pi_{\theta^2}^{HP}$ | 0.44 | 0.48 |
| 3P | 3P Dataset | 0.42 | 0.49 |
| | $\pi_{\theta^3}^{HP}$ | 0.44 | 0.47 |
| | $\pi_{\theta^4}^{HP}$ | 0.44 | 0.46 |

For additional experiments and results, please refer to A.5, A.8 and A.6. We conducted an ablation study by varying the strength of the regularisation term to examine its impact on the final policy. Our findings indicate that initializing training from a BC policy without regularisation leads to weak and/or human-incompatible policies. Finally, we provide further behavioural analysis of our human proxies to illustrate their performance and interactions.

## 6. Results of Baseline Methods on AH2AC2

In this section, we evaluate several baselines in the AH2AC2 challenge, informed by previous research (Hu et al., 2021; Carroll et al., 2019; Strouse et al., 2021). **IPPO**: Policy trained in SP, without human data. **BC**: Policy trained only on available human data using supervised learning. **HDR-IPPO**: Policy builds upon the BC reference by incorporating regularised RL. **BR-BC** (Carroll et al., 2019): Policy is trained by embedding the BC policy in the environment, treating its actions as a part of environment dynamics during

SP learning. **OBL** (Hu et al., 2021): Strong approach for ZSC that does not utilise any human data during training. We use this agent only in the two-player setting since we do not have access to three-player weights. **OP** (Hu et al., 2020): A ZSC method that prevents agents from learning equivalent but mutually incompatible policies across independent training runs. OP accomplishes this by enforcing the equivariance of the policies under the symmetries of the Dec-POMDP, which must be provided as an input of the algorithm. **FCP** (Strouse et al., 2021): A method trained as the best response to a population of self-play agents and their past checkpoints taken throughout training. **DeepSeek-R1** (DeepSeek-AI et al., 2025): A strong reasoning LLM. We benchmark its performance by prompting it to make decisions in Hanabi, without any Hanabi-specific fine-tuning.

We adopt the IPPO configuration from Rutherford et al. (2023) and adapt the BR-BC training procedure from Carroll et al. (2019). These baseline approaches utilise feed-forward architectures. Our BC and HDR-IPPO baselines, in contrast, employ an LSTM-based architecture. While based on the same backbone as the human proxy agents, we use different hyperparameters optimised for the data-limited challenge settings. Additionally, we utilise different random seeds than those used for training the human proxies.

For each of the BC, BR-BC, and HDR-IPPO baselines, we train with three different random seeds. The best agent, based on cross-entropy loss on the validation set, is selected for evaluation. Notably, we observe minimal performance variance across different seeds on the validation set (see Appendix A.3). The IPPO baseline, intended to showcase the limitations of SP in ad-hoc coordination, is trained with a single seed. Finally, we use pre-trained weights and respective hyperparameters for OBL (Hu et al., 2021).

The training population for the FCP agent comprises 36 random seeds. Since Hanabi presents a greater challenge than Overcooked, the environment in which FCP was initially introduced, we employ four checkpoints for each agent. Consequently, the training process for a single FCP agent utilizes 144 checkpoints. Due to the computational demands involved, we train the FCP agent with a single seed.

For DeepSeek-R1, we evaluate its capability using two prompting variants: one providing the LLM solely with the current game state in natural language, and another that additionally includes a description of H-conventions. Further details on these prompts are available in Appendix A.10.

Table 5 shows the initial AH2AC2 leaderboard. OBL (L4) achieves the highest performance without using any human data. In two-player settings, BR-BC achieves the highest score, while HDR-IPPO leads in three-player settings (although OBL lacks pre-trained weights for three-players,

inhibiting its evaluation in this setting). The BC and IPPO baselines perform poorly in both settings, as expected.

These results highlight OBL's effectiveness, surpassing BC, BR-BC, and HDR-IPPO without relying on human data. Methods like OP fail to achieve successful human-AI coordination, and current approaches that leverage limited human data underperform compared to state-of-the-art ZSC algorithms like OBL. This reveals a research gap; existing methods cannot effectively integrate small human datasets to enhance coordination. Additionally, our evaluation of FCP in Hanabi shows it struggles in complex, partially observable environments, suggesting that population-based methods may not provide robust coordination capabilities. There is thus a need for new techniques that efficiently utilise limited human data to improve human-AI teamwork. Although DeepSeek-R1 demonstrates foundational capability, significant improvements are still needed. In the two-player setting, even when prompted with H-conventions, it significantly underperforms compared to OBL. While providing H-conventions substantially improved its score over the basic prompt (5.43 vs 9.91), this approach still falls considerably short of OBL. In the three-player setting, it shows relatively stronger performance, outperforming all other baselines. Our results suggest that while current LLMs, even without fine-tuning, exhibit some inherent coordination capabilities, they do not yet achieve the desired level of efficacy on AH2AC2. Also, it is crucial to note that, due to resource and time constraints, our evaluation of DeepSeek-R1 was conducted on only 100 evaluation games in contrast to the 1000 games used for all other methods, and we leave the action prediction challenge for future work.

## 7. Conclusion

We introduced the Ad-Hoc Human-AI Coordination Challenge (AH2AC2), evaluating human-AI ad-hoc teamplay in the context of the cooperative card game Hanabi. By leveraging a large-scale human play dataset to generate human-like agent policies, we provide a meaningful evaluation framework for assessing AI agents' ability to coordinate with human partners. We released a comprehensive set of baselines, including agents trained with and without human data, to provide a reference point for evaluating novel approaches. Our results highlight the inherent challenge of human-AI coordination. We believe that AH2AC2 represents a significant step forward in the field of human-AI coordination. By providing a standardised evaluation protocol, human proxy agents, and a diverse set of baselines, we aim to foster further research and development. We invite researchers and practitioners alike to participate, pushing the boundaries of what's possible in human-AI collaboration.

Several open challenges and questions remain. We highlight some promising directions for future research:

- **Theoretical Analysis of HDR-IPPO:** While our experiments and previous works provide strong empirical evidence for the effectiveness of regularised RL in generating human-like agents, a deeper theoretical understanding of the methodology is crucial.
- **Generalisation of the benchmark** We currently only cover 2 and 3 players as well as "standard" Hanabi, ignoring the large set of possible variations of the game that are created e.g. by the "rainbow cards". Extending the benchmark to cover those scenarios would be a great way to measure the generalisation ability of agentic systems.
- **Direct Human-AI Play with Human Proxy Agents:** The ultimate validation of our human proxy agents requires direct human-AI play. Future work should involve conducting play experiments with human participants, comparing their experiences and performance when playing with human proxies versus playing with other humans.
- **Comprehensive Evaluation and Advancement of Agentic LLMs:** Our preliminary evaluation of DeepSeek-R1, though limited by resource constraints to fewer games than other baselines, provides initial insights into LLM ca-

pabilities for human-AI coordination. Future work should conduct a more extensive evaluation of LLMs and explore more sophisticated methods. Excitingly, when paired with our human proxies, Hanabi becomes an excellent benchmark for assessing theory of mind in LLMs and their ability to cooperate with humans in complex, partially observable tasks.

*Table 5.* The AH2AC2 leaderboard includes Mean and Median scores achieved for human-AI coordination evaluation and Cross-Entropy Loss (CE) for action prediction challenge. OBL does not make use of available human data, yet it achieves high scores. There is a pressing need for new techniques that efficiently utilise limited human data to improve human-AI coordination.

| Players | Method | Mean | Median | CE |
|---|---|---|---|---|
| | OBL (L4) | **21.04** | 22 | 1.33 |
| | BR-BC | 19.41 | 20 | 10.82 |
| | FCP | 14.01 | 16 | 3.52 |
| **2P** | OP | 13.91 | 19 | 7.81 |
| | HDR-IPPO | 12.76 | 15 | 0.96 |
| | IPPO | 10.16 | 14 | 12.60 |
| | DeepSeek-R1 H-Group | 9.91 | 0 | - |
| | DeepSeek-R1 | 5.43 | 0 | - |
| | BC | 2.12 | 0 | 0.86 |
| | *Human Proxies* [†] | 22.76 | 23 | 0.54 |
| | *BR-BC*\*[†] | 22.59 | 23 | 5.00 |
| | DeepSeek-R1 H-Group | **14.62** | 18 | - |
| | DeepSeek-R1 | 14.38 | 18 | - |
| **3P** | HDR-IPPO | 14.03 | 16 | 0.80 |
| | OP | 12.87 | 18 | 6.40 |
| | BR-BC | 11.89 | 12 | 29.89 |
| | FCP | 11.55 | 6 | 5.97 |
| | IPPO | 6.34 | 0 | 8.60 |
| | BC | 3.31 | 0 | 0.70 |
| | *Human Proxies* [†] | 20.86 | 21 | 0.62 |
| | *BR-BC*\*[†] | 18.80 | 19 | 7.53 |

[†] Not constrained by game limits, acts as a golden standard. BR-BC\* is trained with a BC policy trained on the entire dataset. We report average performance over two human proxies.

## Acknowledgements

We want to thank Johannes Forkel and Darius Muglich for their contributions to various aspects of our work. We also thank Ivan Zubak for helping us set up the website for AH2AC2. AC acknowledges funding from the UKRI AI World Leading Researcher Fellowship (grant EP/W002949/1). AC and JF also acknowledge support from a J.P. Morgan Chase Faculty Research Award. RH gratefully acknowledges support from the Autonomous Intelligent Machines and Systems (AIMS) programme at the University of Oxford, in conjunction with sponsorship from Rosebud.ai. AL acknowledges funding from the Fonds de recherche du Quebec.

## Impact Statement

This paper presents work whose goal is to advance the field of Machine Learning. There are many potential societal consequences of our work, none of which we feel must be specifically highlighted here.

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

# A. Appendix

## A.1. Hanabi

In Hanabi, players aim to collectively build five ascending stacks of cards, one for each of the five colours. The deck comprises 50 cards, with 10 cards of each colour. These include three cards of rank 1, two each of ranks 2, 3, and 4, and a single card of rank 5. A unique feature of the game is that players can see each other's cards, but not their own. Information about one's own cards is gained through hints from teammates or by interpreting their actions. The team starts with eight information tokens and three lives. On their turn, players can choose one of the following three actions: (1) A player can choose a teammate and provide information about all the cards of a specific colour or rank in that teammate's hand. The hint must be complete, meaning all cards of the specified colour or rank must be indicated. Each hint consumes one of the limited information tokens. If no information tokens remain, this action is unavailable. (2) If the team has fewer than eight information tokens, a player can discard a card face-up, revealing it to all players. This action replenishes one information token and the player draws a new card. (3) A player can attempt to play a card from their hand onto the corresponding colour stack. If the card's rank is the next in the sequence for that colour, it is successfully played. Successfully playing a rank-five card earns the team an additional information token. However, an incorrect play results in the loss of one of the team's limited lives. Regardless of the outcome, the player draws a new card from the deck. The game concludes when one of the following three conditions is met: (1) the team loses all their lives, (2) all five stacks are completed, or (3) the deck is depleted. The final score reflects the team's success in building the stacks. The maximum score is 25, and it is determined by summing up all the played cards. When all lives are lost, the score is zero.

## A.2. Further Background

### INDEPENDENT PPO

Proximal Policy Optimisation (PPO) (Schulman et al., 2017) is a method initially developed for single-agent RL. PPO aims to address performance collapse in policy gradient methods. It does this by bounding the ratio of action probabilities between the old and new policies. PPO optimises the objective function,

$$\mathbb{E}_{s \sim d^\pi, a \sim \pi} \left[ \min \left( \frac{\tilde{\pi}(a \mid s)}{\pi(a \mid s)} A^\pi(s, a), \text{clip} \left( \frac{\tilde{\pi}(a \mid s)}{\pi(a \mid s)}, 1 - \epsilon, 1 + \epsilon \right) A^\pi(s, a) \right) \right], \quad (3)$$

where $\text{clip}(t, a, b)$ is a function that outputs $a$ if $t < a$, $b$ if $t > b$ and $t$ otherwise. We consider the extension to multi-agent setting by using independent learning (de Witt et al., 2020). Here, each agent treats the others as part of the environment and learns a critic using its local AOH.

### OBL: ZERO-SHOT COORDINATION

Most SP methods exhibit a tendency towards brittleness and over-coordination, leading to suboptimal performance when paired with independently trained policies. To address this, OBL (Hu et al., 2021) was proposed as an algorithm that learns optimal grounded policies without relying on arbitrary conventions or assumptions about other agents. OBL has demonstrated remarkable success in ZSC scenarios, making it a compelling baseline for ad-hoc teamwork, even though it does not utilise any of the human data provided for the challenge.

## A.3. Additional Results

We provide additional results for performance on the test sets in Table 6 and 7.

## A.4. Training Details

### DATA SPLIT

We begin with a large-scale dataset of game trajectories containing both two-player (2p) and three-player (3p) settings. From this dataset, we set aside validation and test sets. Specifically, for the 2p setting, we select 858 games for validation and 858 for testing; for the 3p setting, we allocate 221 games for validation and 221 for testing.

Once validation and test sets are set aside, the remaining data is used for two purposes: (1) training our human proxy models, and (2) sampling a subset of 1,000 games for each setting (2p and 3p) that we open-source to the research community. Additionally, we open-source our previously defined validation. The test sets, however, are kept closed-source

*Table 6.* Cross-entropy loss on the test set for BC, BR-BC and HDR-IPPO agents in the two-player setting. For human proxies we report results over two available agents. For the rest, we report results over three different seeds. Even though agents are trained with different seeds, they achieve almost identical results. We report loss $\pm SE$.

| Data Used | Method | Test Set | |
|---|---|---|---|
| | | **Loss** | **Accuracy** |
| **Open Sourced Games** | BC | $0.87 \pm 0.0$ | $0.46 \pm 0.0$ |
| | BR-BC | $10.96 \pm 0.1$ | $0.25 \pm 0.0$ |
| | HDR-IPPO | $0.97 \pm 0.0$ | $0.40 \pm 0.0$ |
| **Entire Dataset** | BC | $0.48 \pm 0.0$ | $0.67 \pm 0.0$ |
| | Human Proxy | $0.54 \pm 0.0$ | $0.63 \pm 0.0$ |

*Table 7.* Cross-entropy loss on the test set for BC, BR-BC and HDR-IPPO agents in the three-player setting. For human proxies we report results over two available agents. For the rest, we report results over three different seeds. Even though agents are trained with different seeds, they achieve almost identical results. We report loss $\pm SE$.

| Data Used | Method | Test Set | |
|---|---|---|---|
| | | **Loss** | **Accuracy** |
| **Open Sourced Games** | BC | $0.70 \pm 0.0$ | $0.40 \pm 0.0$ |
| | BR-BC | $32.48 \pm 1.6$ | $0.07 \pm 0.0$ |
| | HDR-IPPO | $0.81 \pm 0.0$ | $0.31 \pm 0.0$ |
| **Entire Dataset** | BC | $0.54 \pm 0.0$ | $0.51 \pm 0.0$ |
| | Human Proxy | $0.62 \pm 0.0$ | $0.44 \pm 0.0$ |

to prevent adaptation or overfitting during model development; these serve exclusively for final performance evaluation. The participants of the challenge are not required to use the same data split as defined here, but are encouraged to do so.

BEHAVIOURAL CLONING (BC)

For each sampled mini-batch of games, we extract individual player trajectories, effectively decomposing each game into $n$ trajectories, where $n$ represents the number of players. This yields sets of trajectories, denoted by $U_d = \{u_0, ..., u_n\}$, with one trajectory per player for each game. Each trajectory, $u_i = \{(o_t^i, a_t^i)\}_{t=1}^T$, consists of local player observations, $o_t^i \in \mathcal{O}^i$, and corresponding actions, $a_t^i \in \mathcal{A}^i$, taken at timestep $t$. Our BC model takes a single player's trajectory as input and predicts an action for each timestep, conditioned on the AOH. A mini-batch of size $b$ is constructed by concatenating multiple sets of trajectories: $\mathcal{D}_{\text{batch}} = U_0 \parallel ... \parallel U_b$, where $\parallel$ denotes concatenation. To augment the training data and improve generalisation, we randomly shuffle the colour space for both observations and actions within each mini-batch before feeding it to the network, as done in the previous works to enhance performance of our model (Hu et al., 2021).

The loss for each batch is calculated as

$$\mathcal{L}^{BC}(\theta) = -\frac{1}{\sum_{\tau_t^i \in \mathcal{D}_{batch}} |\tau_t^i|} \sum_{\tau_t^i \in \mathcal{D}_{batch}} \sum_{t'=1}^{t+1} \ell(\pi_\theta^{BC}(a_{t-1}^i | \tau_{t-1}^i), a_{t-1}^i), \qquad (4)$$

where $\mathcal{D}_{batch}$ is the batch of trajectories $\tau_i$ and $\ell$ is the standard cross-entropy loss. The policy, $\pi_\theta^{BC}$, is conditioned on both the hidden state $\phi_t^i$, encoding the observation history, and the current local observation $o_t^i$.

BEST RESPONSE TO BEHAVIOURAL CLONING (BR-BC)

When training with an embedded human model, we initially train in SP and anneal the amount of SP linearly to zero. Then, we continue training with the BC policy, as Carroll et al. (2019) find that this improves agents' performance. Additionally, we consider the three agent setting that was not considered by Carroll et al. (2019). When annealing, based on the annealing factor, we sample agents to decide whether we train in SP, with a single BC policy or with two BC policies.

IPPO

We follow the same methodology and architecture as presented by (Rutherford et al., 2023).

### A.4.1. OTHER-PLAY (OP)

We employ feed-forward architecture and follow the methodology presented in the original work (Hu et al., 2020).

FICTITIOUS CO-PLAY (FCP)

FCP population consists of IPPO agents, trained with the same methodology as the IPPO baseline. We train a total of 36 agents. For each trained agent, we consider four checkpoints when building a population to train a best response. The final FCP partner agent is modelled using a GRU (Cho et al., 2014), using the same parameters and methodology presented in (Rutherford et al., 2023).

### A.5. Human Proxies: Additional Details

DATASET

Dataset used for training human proxies contains 147,621 Hanabi games, comprising 101,096 two-player and 46,525 three-player games. Table 8 summarises key statistics for scores and game lengths across both player configurations.

*Table 8.* The dataset used for training human proxies comprises a total of 147,621 games, including 101,096 two-player and 46,525 three-player games. The table presents descriptive statistics for game scores and lengths across both player configurations.

| Setting | Metric | Min | Max | Avg | Median | Std |
|---|---|---|---|---|---|---|
| Two-Player | Scores | 1 | 25 | 23.09 | 24.00 | 2.16 |
| | Game Lengths | 2 | 88 | 65.70 | 66.00 | 3.63 |
| Three-Player | Scores | 2 | 25 | 22.94 | 23.00 | 2.10 |
| | Game Lengths | 34 | 78 | 58.38 | 59.00 | 3.36 |

ARCHITECTURES

The architectures and hyperparameters used to train human proxy agents are shown in Table 9. Each model includes a fully connected layer, a multi-layer LSTM block, and a decoder fully connected network that maps the LSTM encodings to a probability distribution over actions. In our implementation, we employ loss masking to ensure that the probability of selecting an illegal action is effectively set to zero during sampling from the policy. Optimisation is performed using the Adam optimiser (Kingma & Ba, 2014) across all configurations. When applying a linear learning rate schedule, the learning rate is reduced to its minimum allowable value during the final 10% of the training process.

HYPERPARAMETER SEARCH

We run a hyperparameter search for the BC policies used for training human proxies. We performed a full grid search across a range of hyperparameters, considering both two-player and three-player game settings. The hyperparameter search space is shown in Table 10.

From these configurations, we selected the two best-performing settings for both two-player and three-player scenarios. These configurations were subsequently employed in the second step of the HDR-IPPO procedure. Respective agents are what we refer to as human proxies.

ROLE OF REGULARISATION

Our human proxy policies have demonstrated successful coordination in cross-play with both other human proxies and the original BC policies. Here, we show that arbitrary policies fail to coordinate well with them. We introduce a new set of agents trained using the same HDR-IPPO procedure as our human proxies, but with a crucial difference, the human data regularisation weight is set to zero, $\lambda = 0$.

*Table 9.* Human proxy agent training configurations and architectures. We showcase both BC and IPPO hyperparameters in a single table.

| Hyperparameter | $\pi_{\theta 1}^{HP}$ | $\pi_{\theta 2}^{HP}$ | $\pi_{\theta 3}^{HP}$ | $\pi_{\theta 4}^{HP}$ |
|---|---|---|---|---|
| **Network Architecture** | | | | |
| Num Players | 2 | 2 | 3 | 3 |
| Activation | GELU | GELU | GELU | GELU |
| LSTM Layers | 512, 512, 512 | 512, 512 | 512, 512, 512 | 512, 512 |
| Input Embedding | 1024 | 1024 | 1024 | 1024 |
| Decoder MLP | 512 | 256 | 1024 | 1024 |
| **BC** | | | | |
| **Optimisation** | | | | |
| Batch Size | 256 | 256 | 128 | 256 |
| Dropout | 0.5 | 0.3 | 0.5 | 0.5 |
| LR Schedule | Linear | Linear | Linear | Linear |
| Initial LR | 0.005 | 0.005 | 0.005 | 0.005 |
| Final LR | 5.0e-05 | 5.0e-05 | 1.0e-05 | 1.0e-05 |
| Epochs | 50 | 50 | 70 | 70 |
| **Training** | | | | |
| Permute Colours | Yes | Yes | Yes | Yes |
| Self-Play Eval Games | 5000 | 5000 | 5000 | 5000 |
| **IPPO** | | | | |
| **Optimisation** | | | | |
| Learning Rate | 0.0003 | 0.0003 | 0.0005 | 0.0003 |
| Gamma Discount | 0.99 | 0.99 | 0.99 | 0.99 |
| GAE Lambda | 0.95 | 0.95 | 0.95 | 0.95 |
| Clip Epsilon | 0.2 | 0.2 | 0.2 | 0.2 |
| Entropy Coefficient | 1.0e-05 | 0.0001 | 0.0001 | 0.0001 |
| Value Function Coeff | 0.5 | 0.5 | 0.5 | 0.5 |
| Max Gradient Norm | 0.5 | 0.5 | 0.5 | 0.5 |
| Update Epochs | 4 | 4 | 4 | 4 |
| Num Minibatches | 4 | 4 | 4 | 4 |
| **Critic Network** | | | | |
| Critic MLP | 512 | 512 | 512 | 512 |
| **Training** | | | | |
| BC Policy KL Weight | 0.3 | 0.2 | 0.1 | 0.15 |
| Total Timesteps | 5e9 | 5e9 | 5e9 | 5e9 |
| **Environments** | | | | |
| Num Env Steps | 90 | 90 | 90 | 90 |
| Num Train Envs | 1024 | 1024 | 1024 | 1024 |
| Num Eval Envs | 512 | 512 | 512 | 512 |

*Table 10.* Hyperparameter search space for BC when developing human proxies.

| Hyperparameter | Values |
|---|---|
| Dropout | 0.3, 0.5 |
| Batch Size | 64, 128, 256 |
| LSTM Layers | (512, 512), (512, 512, 512), (1024, 1024), (1024, 1024, 1024) |
| Input Embedding | 512, 1024, (512, 512) |
| Decoder MLP | 512, 1024 |

These new policies start from the same BC policy weights as our human proxies and utilise identical hyperparameters, except for the KL regularisation term, which is eliminated. We focus on two specific agents, one for the two-player setting (based on $\pi_{\theta 2}^{BC}$) and one for the three-player setting (based on $\pi_{\theta 4}^{BC}$). We use the weights obtained at the final training timestep as checkpoints for these model weights.

Firstly, we observed a rapid deterioration in SP performance for the non-regularised policies. The agents quickly lost the ability to perform well, even in SP. This degradation is evident in Figure 3 and Figure 4. This sharp decline suggests a divergence from the strategies initially learned during the BC phase. Without the regularisation term guiding the policy towards human-like behaviour, the agents appear to explore alternative strategies that may lead to suboptimal performance in the context of the game.

After the initial performance drop, the scores of the non-regularised policies gradually improve during training. However, with the current set of hyperparameters, this progress is slow and we do not observe convergence. The agents seem unable to efficiently re-learn a different, yet still effective, policy. It is important to note that this behaviour is not observed with different hyperparameter settings, but for the purpose of direct comparison in this analysis, we maintained the same configuration as the human proxy agents.

When paired with human proxies and BC policies, the non-regularised agents exhibit significantly lower coordination and overall performance compared to the pairings between human proxies and BC policies exclusively. This discrepancy highlights the crucial role of the human data regularisation term in ensuring that the learned strategies remain aligned with human-like play, facilitating successful coordination.

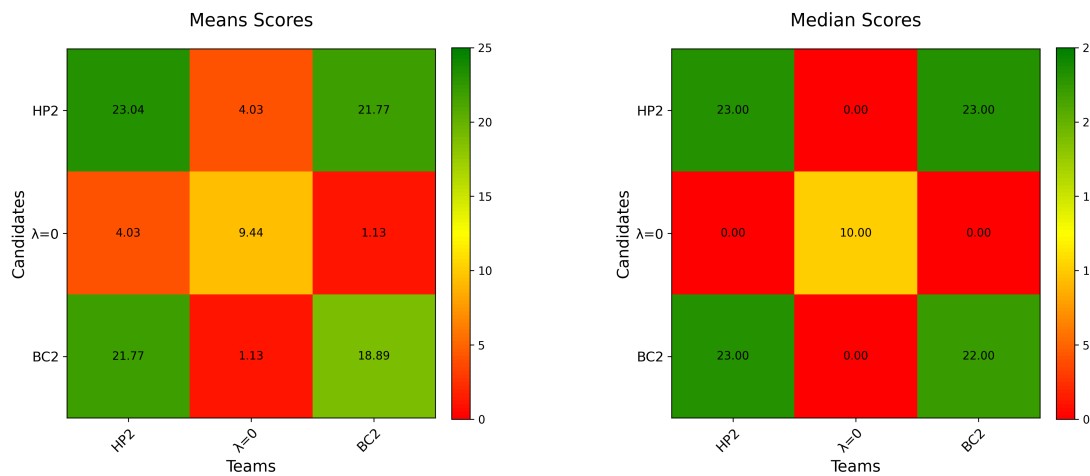

*Figure 3.* Cross-play performance matrix for two-player agents, comparing a human proxy agent ($\pi_{\theta 2}^{HP}$), its corresponding BC policy ($\pi_{\theta 2}^{BC}$), and a non-regularised HDR-IPPO agent initialised from the same BC policy, $\pi_{\theta 2}^{BC}$, but where $\lambda = 0$. Each matrix element represents the average score achieved by a team, averaged over both player orderings. Averages are calculated based on 15,000 games per team permutation.

Consider the three-player cross-play matrix depicted in Figure 4. The BC policy, while achieving a low SP score of 7.19, significantly improves to 18.92 when paired with two human proxies. This substantial boost underscores the BC policy's inherent understanding of human-like strategies, even if it struggles to execute them independently. In contrast, the non-regularised policy, with a SP score of 3.99, only sees a marginal improvement to 6.93 in cross-play with human proxies. Furthermore, pairing the BC policy (the stronger player in SP) with two non-regularised policies actually decreases the performance compared to the SP performance of non-regularised policies. This degradation suggests a fundamental incompatibility between the strategies learned by the non-regularised policy and the human-like conventions given by the BC policy.

## A.6. Behaviour Analysis

The dataset we use in this work comes from the human policies that adhere to H-Group Conventions. This distinguishes our work from previous human-AI coordination studies, as these strategies can be explicitly described in natural language and are documented.

We also include behavioural metrics proposed by (Canaan et al., 2020), for both the trajectories found in the dataset and the ones created by the human proxies. Results are shown in Table 11.

- Information per Play (IPP): This metric assesses how much information the agent possesses about each card it plays. For every card played, we determine the number of known attributes (0, 1, or 2, representing colour and rank). These values are averaged across all played cards. The final score is then normalized by dividing by 2, resulting in a range between 0 and 1. An agent with an IPP of 1 exclusively plays cards for which it knows both the colour and rank. Conversely, an agent with an IPP of 0 only plays cards about which it has no prior knowledge.

- Communicativeness: This metric quantifies how often an agent chooses to communicate information. It is calculated as

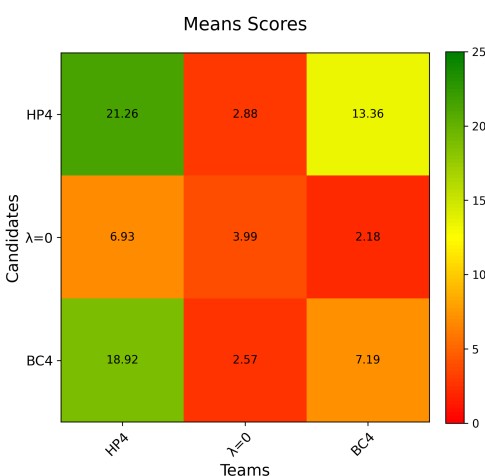
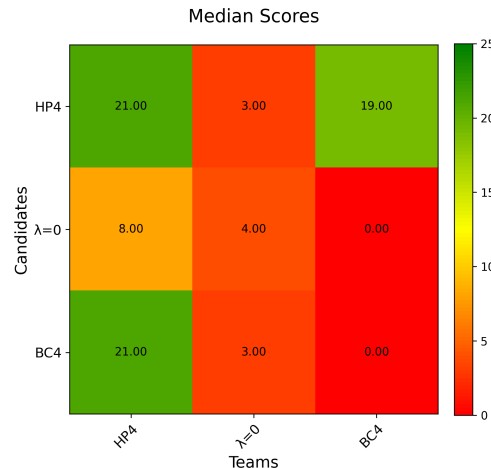

*Figure 4.* Cross-play performance matrix for three-player setting, with scores for a human proxy agent ($\pi_{\theta^4}^{HP}$), its corresponding BC policy ($\pi_{\theta^4}^{BC}$), and a non-regularised HDR-IPPO agent initialised from the same BC policy, $\pi_{\theta^4}^{BC}$, but with $\lambda = 0$. Each matrix element $(i, j)$ represents the average score achieved by a team composed of two instances of one agent on the $x$-axis and one instance of agent on the $y$-axis, averaged over all possible permutations of player positions. Averages are calculated based on 15,000 games per team permutation.

the proportion of turns in which the agent opts to give a hint, given that a hint token is available at the beginning of the turn. A fully communicative agent, scoring 1, would always provide a hint when possible. An agent that never gives hints, scoring 0, would be considered completely non-communicative.

*Table 11.* We compare behaviour features available in the dataset and once present in trajectories generated by our human proxies. Precisely, we compute Information per Play (IPP) and Communicativeness. We show metrics are almost identical for both cases.

| Trajectories | Proxy | IPP | Communicativeness |
|---|---|---|---|
| Two-Player | 2P Dataset | 0.44 | 0.47 |
| | $\pi_{\theta^1}^{HP}$ | 0.43 | 0.45 |
| | $\pi_{\theta^2}^{HP}$ | 0.44 | 0.48 |
| Three-Player | 3P Dataset | 0.42 | 0.49 |
| | $\pi_{\theta^3}^{HP}$ | 0.44 | 0.47 |
| | $\pi_{\theta^4}^{HP}$ | 0.44 | 0.46 |

Furthermore, we provide a qualitative assessment by analysing games played by our human proxies. Our findings show that our agents exhibit highly human-like behaviour and in most instances adhere to H-convention strategies. Sample renders can be found in the anonymous repository we provide and here we analyse the first game provided. We analyse each of the moves below, marking each move as a success or failure, depending on whether the convention is followed correctly. We also find that the mistakes made when not following conventions are very human-like mistakes. For example, on Turn 2, the agent violates the good touch principle, but this move also looks a lot like a 2 save. Hence, it violates a convention because it tried to follow a different one, which was not applicable in this case. Most of the failure cases we found come from confusing conventions that should be played in that particular instance, which is a human-like mistake.

1. Turn 0: Success (Hint blue, play clue on the blue 1)

2. Turn 1: Success (Play clue on both the 1's)

3. Turn 2: Failure (Looks like a 2 save, but doubles the yellow 2's, violating good touch principle)

4. Turn 3: Success (Actor 1 plays card slot 2, knowing it's a B1, which follows the "play clue" convention)

5. Turn 4: Success (Actor 0 gives a 5 save when Red 5 is on the chop)

6. Turn 5: Success (Actor 1 plays a known playable card)

7. Turn 6: Failure (Actor 0 should have 2-saved the Red 2 on chop)

8. Turn 7: Success (Actor 1 discards Red 2, the chop card)

9. Turn 8: Success (Actor 0 plays slot 0, a known playable 1)

10. Turn 9: Success (Actor 1 discards chop on slot 3)

11. Turn 10: Success (Fix clued the duplicate cards)

12. Turn 11: Success (Plays known playable card)

13. Turn 12: Success (Gives play clue to the Red 1)

14. Turn 13: Success (Plays the Red 1)

15. Turn 14: Success (Discards chop)

16. Turn 15: Success (Gives play clue to the Yellow 3)

17. Turn 16: Success (Plays Yellow 3)

18. Turn 17: Success (Discards known trash)

19. Turn 18: Success (Discards chop)

20. Turn 19: Failure (Doesn't understand chop-focus, giving play clue to wrong card, and violates good touch principle by clueing Red 1 and Red 3 twice)

21. Turn 20: Failure (The focus of the last clue was the chop, so it should have played the Red 3, but played Red 2 instead)

22. Turn 21: Success (Fix clue on the duplicated Red 3's)

23. Turn 22: Success (Plays known playable Red 3)

24. Turn 23: Success (Discards chop, position 1)

25. Turn 24: Success (Gives play clue to White 2)

26. Turn 25: Success (Plays White 2)

27. Turn 26: Success (Play clue on the Red 4)

28. Turn 27: Success (Plays the Red 4)

29. Turn 28: Success (Play clue on Blue 4, and filling in White 3)

30. Turn 29: Success (Plays White 3)

31. Turn 30: Success (Discards known trash)

32. Turn 31: Success (Play clue on White 4)

33. Turn 32: Success (Plays White 4)

34. Turn 33: Success (Plays known playable Red 5)

35. Turn 34: Success (Discards known trash Red 1)

36. Turn 35: Failure (Should have played its Blue 3 because of the play clue, instead gave a 5 hint off chop, which is illegal after the late game)

37. Turn 36: Success (Discards chop)

38. Turn 37: Failure (Should have played its Blue 3, instead gave a 2 hint which is illegal)

39. Turn 38: Success (Discards chop)

40. Turn 39: Failure (Should have played its Blue 3, instead discarded chop)

41. Turn 40: Success (Play clue on Blue 4, filling in Blue 3)

42. Turn 41: Success (Plays Blue 3)

43. Turn 42: Success (Discards chop)

44. Turn 43: Success (Plays Blue 4)

45. Turn 44: Success (Discards chop)

46. Turn 45: Success (Hint Blue, filling in Blue 5)

47. Turn 46: Success (Plays Blue 5)

48. Turn 47: Success (Play clue on Green 1)

49. Turn 48: Success (Plays Green 1)

50. Turn 49: Success (Discards chop)

51. Turn 50: Success (Play clue on Yellow 4)

52. Turn 51: Success (Plays Yellow 4)

53. Turn 52: Success (Plays Green 2)

54. Turn 53: Success (Discards chop)

55. Turn 54: Success (Reveals Green 4 identity)

56. Turn 55: Success (Discards chop)

57. Turn 56: Success (Plays Yellow 5)

58. Turn 57: Success (Discards chop)

59. Turn 58: Success (5 save on Green 5)

60. Turn 59: Success (Stalling, hinting 1s)

61. Turn 60: Failure (Hinting Green is seen as a play clue on Green 1, which is illegal)

62. Turn 61: Success (Plays Green 1, which it thought was Green 3 because of convention)

63. Turn 62: Success (Play clue on Green 3)

64. Turn 63: Success (Plays Green 3)

65. Turn 64: Success (Hints White 5)

66. Turn 65: Success (Plays Green 4)

In summary, in this game, the human proxy followed H-group conventions for $88\%$ of the moves and used various strategies while playing the game.

- **Successful H-Group conventions played:**

    - Giving play clue (14x)
    - Responding to play clue (23x)
    - 2 Save (1x)
    - 5 Save (2x)
    - Discard chop (13x)
    - Fix clue (3x)
    - Discard known trash (3x)
    - Stall clue (1x)

- **H-Group Conventions violated:**

    - Violating Good touch Principle (1x)
    - Failure to 2 save (1x)
    - Incorrectly gives chop focus clue (1x)
    - Incorrectly responding to chop focus clue (1x)
    - Failure to play play clue (3x)
    - Incorrectly give play clue (1x)

### A.7. AH2AC2: Challenge Implementation Details

We have established a dedicated website for the AH2AC2 challenge at https://ah2ac2.com/. Participants can request to join the challenge through this website, where they will also find a leaderboard displaying existing results.

Upon submitting a participation request, candidates will be notified when the challenge becomes public and will receive an API key for testing their implementations and for official evaluation. The results of candidate agents will automatically appear on the leaderboard once all games are completed. Participation in the action prediction challenge is currently optional but encouraged.

For the evaluation, we assess the performance of each candidate agent over a total of 2,000 games played with our human proxy agents: 1,000 games in the two-player setting and 1,000 games in the three-player setting. We consider all seating configurations and all combinations of agents and seating positions, including scenarios where candidates control multiple agents (except for self-play situations). The evaluation is conducted uniformly across different seating configurations to ensure a fair and comprehensive assessment. This setup follows the procedure introduced in the seminal work on ad hoc teamwork evaluation by (Stone et al., 2010). Participants can obtain evaluation results and information for their agents at any time through our dedicated API.

#### Code and Data Availability

We provide our code in an anonymous repository: `https://anonymous.4open.science/r/ah2ac2-2FDA`.

At the time of submission, the open-sourced codebase includes:

- Data, including splits, used to train AH2AC baselines.

- Code for training the baselines.

- Pre-trained weights for all provided baselines.

- An API for accessing human proxies.

- Implementations for Human-AI Coordination evaluations for all baselines except FCP, as the FCP implementation is in its final testing phase.

### A.8. HDR-IPPO: Ablation Study

We conduct an ablation study to investigate the impact of the human data regularisation term on the performance and behaviour of agents trained with the HDR-IPPO. By systematically varying the strength of the regularisation, we aim to gain insights into its role in guiding policy learning towards strategies learned during BC (in this case, human-like strategies).

METHODOLOGY

We focus on the two-player setting, where BC has demonstrated strong performance as a baseline. Due to the computational and time constraints associated with training HDR-IPPO agents, it was not feasible to extend this study to the three-player setting. To systematically analyse the effect of the KL regularisation term introduced in the HDR-IPPO algorithm, we adopt the following methodology:

1. We train a new BC policy that serves as both a starting point for subsequent HDR-IPPO training and a baseline for comparison in this ablation study.

2. From the baseline BC policy, we train multiple HDR-IPPO agents. These agents share identical architectures, hyperparameters, and training procedures, with the sole exception of the human data regularisation weight, $\lambda$. We vary the weight of regularisation term across a range of values, from no regularisation ($\lambda = 0.00$) to very high regularisation ($\lambda = 0.70$) to comprehensively investigate the effects of this term during both training and evaluation. The final policy weights for each agent are stored for subsequent analysis.

3. We evaluate, analyse and compare the trained HDR-IPPO agents and the baseline BC policy. Precisely, we:

   (a) Assess the SP performance of each agent to understand how the strength of KL regularisation influences its ability to play Hanabi effectively on its own.
   (b) Evaluate the cross-play performance of each HDR-IPPO agent when paired with the baseline BC policy. Higher scores in this setting indicate that the HDR-IPPO agent has learned strategies compatible with the human-like conventions exhibited by the BC agent.
   (c) Conduct cross-play evaluations among different HDR-IPPO agents to identify potential coordination patterns.
   (d) Evaluate each agent on the held-out validation and test sets of human data. Good performance on this data indicates that the agent has effectively retained the conventions observed in the human demonstrations.
   (e) Plot the KL divergence between the action distributions of each HDR-IPPO agent and the baseline BC policy throughout the training process. This visualisation allows us to identify potential convergence or divergence patterns for the KL divergence term.

EXPERIMENTAL SETUP

The hyperparameters used for training the agents in this ablation study are listed in Table 12.

*Table 12.* Hyperparameters used for training agents in the ablation study. We showcase both BC and IPPO hyperparameters in a single table.

| Hyperparameter | Value |
| --- | --- |
| **Network** | |
| Num Players | 2 |
| Activation | GELU |
| LSTM Layers | 512 |
| Input Embedding | 1024 |
| Decoder MLP | 512 |
| Dropout | 0.5 |
| **BC** | |
| **Optimisation** | |
| Optimiser | Adam (Kingma & Ba, 2014) |
| Batch Size | 128 |
| LR Schedule | Linear |
| Initial LR | 0.005 |
| Final LR | 1.0e-05 |
| Epochs | 70 |
| **Training** | |
| Permute Colours | Yes |
| Self-Play Eval Games | 5000 |
| **IPPO** | |
| **Optimization** | |
| Learning Rate | 0.0005 |
| Gamma Discount | 0.99 |
| GAE Lambda | 0.95 |
| Clip Epsilon | 0.2 |
| Entropy Coefficient | 0.01 |
| Value Function Coeff | 0.5 |
| Max Gradient Norm | 0.5 |
| Update Epochs | 4 |
| Num Minibatches | 4 |
| **Critic Network** | |
| Critic MLP | 512 |
| **Training** | |
| Total Timesteps | 2e10 |
| **Environments** | |
| Num Env Steps | 128 |
| Num Train Envs | 1024 |
| Num Eval Envs | 512 |
| Num Test Actors | 1024 |
| Num Test Envs | 512 |

In addition to the listed parameters, we consider setups with the following human data KL regularisation weights: 0.00, 0.01, 0.08, 0.13, 0.20, 0.30, 0.50, and 0.70. This range allows us to explore the effects of regularisation from a complete absence to a very strong influence on the learning process.

RESULTS AND DISCUSSION

We start by presenting the cross-play matrix in Figure 5.

From the cross-play matrix, we observe consistently high SP scores across all agents, indicating that all variations of HDR-IPPO lead to significant improvements over the baseline BC policy's mean SP score of 19.53. The HDR-IPPO agents with the lowest SP scores are those at the extremes of the regularisation spectrum: the non-regularised policy ($\lambda = 0.00$) and the one with the highest regularisation ($\lambda = 0.70$). The agent showing the best SP performance, with a mean score of 23.42, is the one trained with $\lambda = 0.13$.

This result provides initial empirical evidence supporting the effectiveness of the regularisation employed in HDR-IPPO. We hypothesise that excessively high regularisation weights might prevent the policy to generalise, while the complete absence of regularisation could lead to the adoption of strategies that deviate significantly from human-like conventions.

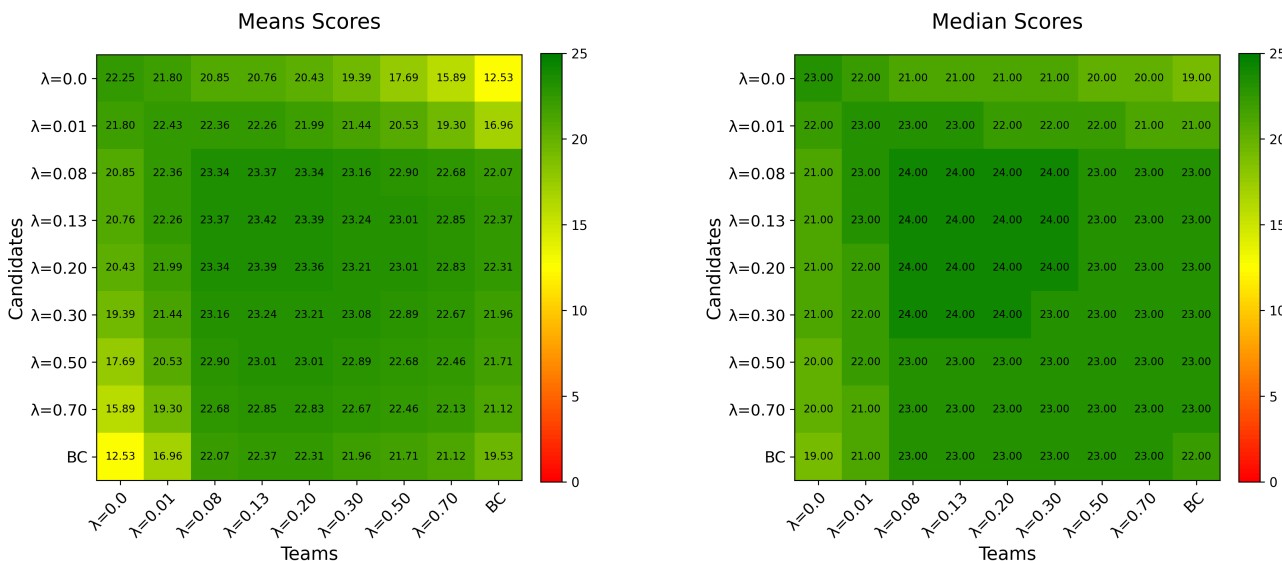

*Figure 5.* Cross-play performance matrix. Each element $(i, j)$ represents the average score achieved by a team comprising of Agent $j$ and Agent $i$, averaged over possible permutations of player positions. The average score for each team is calculated based on 25,000 games per permutation.

Next, we examine the distribution of perfect and zero-score games. As shown in the Table 13 below, with an appropriate level of regularisation, we acquire a policy that achieves a high number of perfect games while completely eliminating games with a score of zero. In contrast, the non-regularised policy ($\lambda = 0.00$) yields only 143 perfect games, and its minimum score is comparable to that of the policy with $\lambda = 0.13$, which boasts 1599 perfect games. This discrepancy further supports our hypothesis that these agents employ fundamentally different strategies.

*Table 13.* Perfect and zero-score games count out of 5,000 SP games. We also show the minimum score in the set of 5,000 games.

|                     | **Perfect** | **Zero** | **Min** |
| ------------------- | ----------- | -------- | ------- |
| $\lambda = 0.00$    | 143         | 0        | 13      |
| $\lambda = 0.01$    | 247         | 0        | 14      |
| $\lambda = 0.08$    | 1351        | 0        | 14      |
| $\lambda = 0.13$    | 1599        | 0        | 13      |
| $\lambda = 0.20$    | 1821        | 0        | 15      |
| $\lambda = 0.30$    | 1564        | 3        | 0       |
| $\lambda = 0.50$    | 1280        | 21       | 0       |
| $\lambda = 0.70$    | 1166        | 113      | 0       |

Let us now analyse the cross-play results from the perspective of the BC policy. Examining the BC policy column in Figure 5, we observe that cross-play scores generally increase compared to the initial BC SP score when paired with most HDR-IPPO agents. However, a notable exception occurs for pairings with agents trained using $\lambda = 0.00$ and $\lambda = 0.01$. Despite these two policies having significantly higher SP scores than the BC policy, their coordination with the BC agent is poor, resulting in a substantial drop in cross-play performance. This widening gap between SP and cross-play scores is a recognised indicator of poor coordination. Additionally, the policy with the highest regularisation weight ($\lambda = 0.70$) does not exhibit this coordination breakdown, even though it performs worse in SP compared to the non-regularised agent ($\lambda = 0.00$). This observation provides empirical evidence that training with insufficient regularisation can lead to a divergence from the strategies encountered in the dataset used for BC, even when the final policy develops strong strategies and performs well in isolation.

Furthermore, let's examine the cross-play results from the perspective of the non-regularised policy ($\lambda = 0.00$). A clear trend emerges where cross-play performance deteriorates as the regularisation weight of the partner policy increases. This

suggests that the non-regularised policy, having diverged from human-like conventions, struggles to coordinate effectively with agents that adhere more closely to those conventions. In contrast, for policies trained with higher regularisation weights, we observe the opposite trend. The gap between SP and cross-play scores diminishes, and in some cases, cross-play even yields higher scores than the weaker policy's SP performance. This indicates that these policies, guided by the KL regularisation term, converge towards similar strategies, enabling them to coordinate exceptionally well, even surpassing individual SP scores in certain instances. The same holds true when pairing these agents with the baseline BC agent, further underscoring their compatibility with human-like conventions. This analysis provides strong empirical evidence that policies trained with higher regularisation weights tend to converge to a shared set of strategies, and that those strategies align closely with ones learned during the BC procedure.

We now shift our focus to evaluating the agents on the held-out validation and test sets, as presented in Table 14. Increasing the regularisation weight generally leads to improved performance on the held-out data, suggesting better adherence to the conventions present in the training set. Notably, the increase from $\lambda = 0.00$ to $\lambda = 0.08$ results in a substantial improvement of over 20% accuracy on both the validation and test sets. Importantly, subsequent increases yield only marginal gains.

Importantly, agents that perform well in cross-play tend to have a lot stronger performance on the held-out datasets. Hence, we again show that they converge to similar conventions, but we also show that these conventions are very close to ones learned during training.

*Table 14.* Performance on hold out sets for all agents. We show the best result in **bold**.

|  | **Val Loss** | **Val Acc** | **Test Loss** | **Test Acc** |
| --- | --- | --- | --- | --- |
| BC | 0.466 | **0.676** | **0.468** | **0.674** |
| $\lambda = 0.00$ | 7.368 | 0.330 | 7.385 | 0.327 |
| $\lambda = 0.01$ | 1.354 | 0.435 | 1.369 | 0.433 |
| $\lambda = 0.08$ | 0.716 | 0.574 | 0.725 | 0.571 |
| $\lambda = 0.13$ | 0.618 | 0.607 | 0.628 | 0.605 |
| $\lambda = 0.20$ | 0.540 | 0.636 | 0.548 | 0.634 |
| $\lambda = 0.30$ | 0.488 | 0.659 | 0.495 | 0.656 |
| $\lambda = 0.50$ | 0.475 | 0.665 | 0.481 | 0.663 |
| $\lambda = 0.70$ | **0.464** | 0.671 | 0.469 | 0.668 |

Finally, we turn our attention to the evolution of the KL divergence term throughout the training process. Figure 6 illustrates the KL divergence for all trained HDR-IPPO policies. It is immediately apparent that the KL terms for $\lambda = 0.00$ and $\lambda = 0.01$ are significantly higher than the others, rendering the remaining curves barely visible in the plot. This observation aligns with our previous findings, further reinforcing the notion that insufficient regularisation can lead to substantial divergence from the human-like strategies learned through BC.

To gain a clearer understanding of the KL divergence dynamics for policies with higher regularisation weights, we present Figure 7, which excludes the policies with $\lambda = 0.00$ and $\lambda = 0.01$. While policies with lower regularisation weights show increasing KL divergence during training, those with higher weights demonstrate a decreasing trend. This suggests that stronger regularisation effectively prevents the policy from deviating too far from the human-like strategies captured by the BC policy.

We do observe an initial increase in KL divergence for all policies during the first few update steps. This is likely attributable to the dominance of the IPPO loss over the KL divergence term in the early stages of training, particularly as the value function is being learned from scratch.

In conclusion, our ablation study provides compelling evidence that the KL regularisation term in HDR-IPPO effectively prevents divergence from the human-like strategies learned through BC. However, it is crucial to set the regularisation weight, $\lambda$, to a sufficiently large value to ensure sustained adherence to these conventions throughout the training process, particularly in the context of extended training duration.

### A.9. Baselines: Hyperparameters

For OBL, we use open-sourced weights and hyperparameters (Hu et al., 2021).

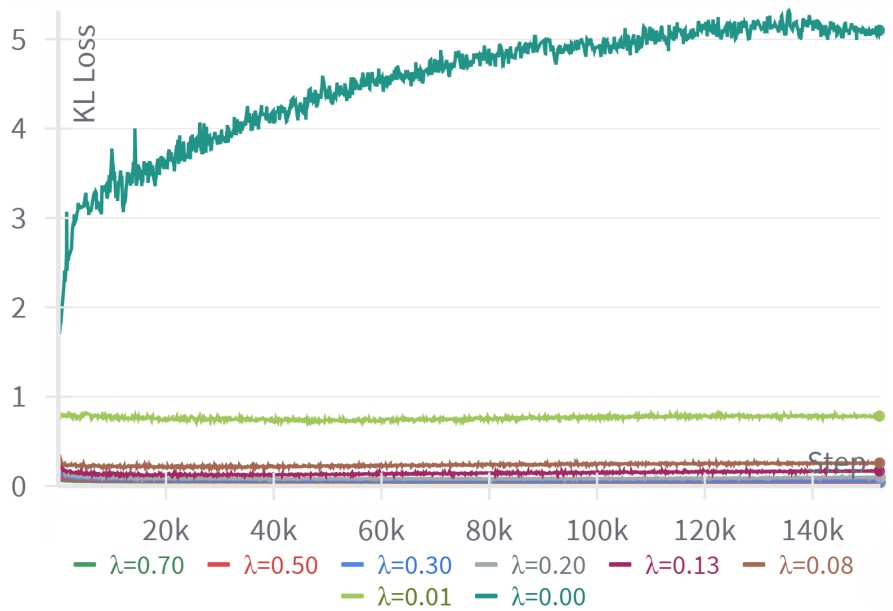

*Figure 6.* KL divergence throughout training for all trained HDR-IPPO policies.

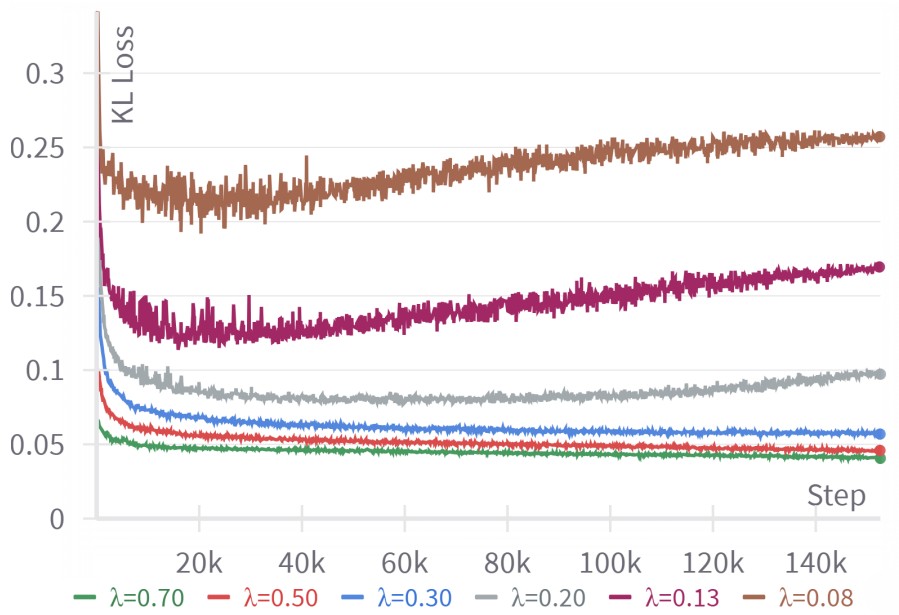

*Figure 7.* KL divergence throughout training where $\lambda \geq 0.08$.

*Table 15.* Hyperparameters used for training BC, HDR-IPPO baselines on a 1,000-game data limit challenge. BC policies trained as baselines are used for starting points in HDR-IPPO and later for BR-BC.

| Hyperparameter | Two-Player Setting | Three-Player Setting |
|---|---|---|
| **Network Architecture** | | |
| Num Players | 2 | 3 |
| Activation | GELU | GELU |
| LSTM Layers | 512 | 512 |
| Input Embedding | 1024 | 512 |
| Decoder MLP | 256 | 256 |
| **BC** | | |
| **Optimisation** | | |
| Batch Size | 32 | 128 |
| Dropout | 0.5 | 0.5 |
| LR Schedule | Linear | Linear |
| Initial LR | 0.005 | 0.005 |
| Final LR | 0.0001 | 0.0001 |
| Epochs | 70 | 50 |
| **Training** | | |
| Permute Colours | Yes | Yes |
| Self-Play Eval Games | 5000 | 5000 |
| **IPPO during HDR-IPPO** | | |
| **Optimisation** | | |
| Learning Rate | 0.0005 | 0.0005 |
| Linear Schedule | True | True |
| Gamma Discount | 0.99 | 0.99 |
| GAE Lambda | 0.95 | 0.95 |
| Clip Epsilon | 0.2 | 0.2 |
| Entropy Coefficient | 0.001 | 0.001 |
| Value Function Coeff | 0.5 | 0.5 |
| Max Gradient Norm | 0.5 | 0.5 |
| Update Epochs | 4 | 4 |
| Num Minibatches | 4 | 4 |
| **Critic Network** | | |
| Critic MLP | 512 | 512 |
| **Training** | | |
| BC Policy KL Weight | 0.25 | 0.25 |
| Total Timesteps | 1e10 | 1e10 |
| **Environments** | | |
| Num Env Steps | 128 | 128 |
| Num Train Envs | 1024 | 1024 |
| Num Eval Envs | 512 | 512 |

*Table 16.* Hyperparameters used for training BC, HDR-IPPO baselines on a 5,000-game data limit challenge. BC policies trained as baselines are used for starting points in HDR-IPPO and later for BR-BC.

| Hyperparameter | Two-Player Setting | Three-Player Setting |
|---|---|---|
| **Network Architecture** | | |
| Num Players | 2 | 3 |
| Activation | GELU | GELU |
| LSTM Layers | 512 | 512 |
| Input Embedding | 1024 | 512 |
| Decoder MLP | 256 | 256 |
| **BC** | | |
| **Optimisation** | | |
| Batch Size | 128 | 256 |
| Dropout | 0.5 | 0.5 |
| LR Schedule | Linear | Linear |
| Initial LR | 0.005 | 0.005 |
| Final LR | 0.0001 | 0.0001 |
| Epochs | 50 | 50 |
| **Training** | | |
| Permute Colours | Yes | Yes |
| Self-Play Eval Games | 5000 | 5000 |
| **IPPO during HDR-IPPO** | | |
| **Optimisation** | | |
| Learning Rate | 0.0005 | 0.0005 |
| Linear Schedule | True | True |
| Gamma Discount | 0.99 | 0.99 |
| GAE Lambda | 0.95 | 0.95 |
| Clip Epsilon | 0.2 | 0.2 |
| Entropy Coefficient | 0.001 | 0.001 |
| Value Function Coeff | 0.5 | 0.5 |
| Max Gradient Norm | 0.5 | 0.5 |
| Update Epochs | 4 | 4 |
| Num Minibatches | 4 | 4 |
| **Critic Network** | | |
| Critic MLP | 512 | 512 |
| **Training** | | |
| BC Policy KL Weight | 0.25 | 0.25 |
| Total Timesteps | 1e10 | 1e10 |
| **Environments** | | |
| Num Env Steps | 128 | 128 |
| Num Train Envs | 1024 | 1024 |
| Num Eval Envs | 512 | 512 |

*Table 17.* Hyperparameters used for training all IPPO and BR-BC baseline agents. Here, we use feed-forward architecture. BC agents used for BR-BC are the same as shown in Table 15 and 16, depending on the challenge variety.

| Hyperparameter | Two-Player Setting | Three-Player Setting |
|---|---|---|
| **Network Architecture** | | |
| Num Players | 2 | 3 |
| Activation | RELU | RELU |
| MLP | (512, 512) | (512, 512) |
| **IPPO** | | |
| **Optimisation** | | |
| Learning Rate | 0.0005 | 0.0005 |
| Gamma Discount | 0.99 | 0.99 |
| GAE Lambda | 0.95 | 0.95 |
| Clip Epsilon | 0.2 | 0.2 |
| Entropy Coefficient | 0.01 | 0.01 |
| Value Function Coeff | 0.5 | 0.5 |
| Max Gradient Norm | 0.5 | 0.5 |
| Update Epochs | 4 | 4 |
| Num Minibatches | 4 | 4 |
| **Critic Network** | | |
| Critic MLP | 512 | 512 |
| **Training** | | |
| Total Timesteps | 1e10 | 1e10 |
| **Environments** | | |
| Num Env Steps | 128 | 128 |
| Num Train Envs | 1024 | 1024 |
| **BR-BC** | | |
| BC Anneal Start | 1e9 | 1e9 |
| BC Anneal End | 6e9 | 6e9 |

## A.10. LLM Prompts: Additional Details

To benchmark off-the-shelf large language models (LLMs) in cooperative play, we transform every Hanabi observation into a natural-language prompt and perform zero-shot inference, i.e., without parameter updates or retrieval augmentation (Lewis et al., 2020). Modern work on chain-of-thought prompting demonstrates that LLMs reason more reliably when instructions are explicit and structured (Wei et al., 2022). Conversely, long uncurated contexts increase the risk of hallucinations—especially the "lost-in-the-middle" error in which models ignore mid-prompt evidence (Liu et al., 2023). We therefore send the model only (i) a Hanabi rules prompt segment, (ii) the H-Group conventions, and (iii) the observation for the current game state. These prompt segments are populated from parameterised templates so that any variant (player count, token budget, deck composition) can be generated dynamically.

### A.10.1. HANABI RULES PROMPT

This prompt segment embeds a complete description of the Hanabi rules. It specifies deck size, token mechanics, legal actions, hand indexing, and end-game triggers. All imperative verbs match the action names used later in the "Valid Actions" list, so a parser can easily identify the LLM's choice:

```
You are playing the card game Hanabi, the three player variant. Every turn I will
give you the state of the game, the actions that your teammates just took, your
teammates hands, and your hand. You will then need to choose an action to take
based on the state of the game.

# Hanabi Game Rules

## Overview and Objective:
- Hanabi is a cooperative card game in which players take turns working together
  to build firework card sequences in five colors.
- Each sequence must begin with a rank 1 card and continue in increasing order up
  to rank 5.
- The goal is to complete as many fireworks stacks as possible; a perfect game
  scores 25 when every color stack is completed.

## Game Components:
- Deck: The deck consists of cards in five colors (Red, Yellow, Green, White,
  Blue). For each color there are three copies of rank 1, two copies of rank 2,
  two copies of rank 3, two copies of rank 4, and only one copy of the rank 5
  card.
- Clue Tokens: There are 3 clue tokens available. These are spent when giving
  clues and can be regained by discarding a card or playing a 5.
- Life Tokens: There are 8 life tokens available. Each misplayed card causes the
  loss of one life token; if all three are lost, the game ends immediately, and
  all players lose.
- Firework Stacks: There is one stack for each color. Cards are added to these
  stacks in ascending order.
- Discard Pile: A common area where discarded or misplayed cards are placed that
  all players can see.
- Player Hands: Each player's hand is arranged so that other players can see the
  cards, but the owner cannot see their own cards. So you can't see the identies
  of your own cards, but you know the identies of your teammates cards.
- Deck Draw Pile: The remaining deck from which new cards are drawn.

## Game Turn and Actions:
- On each turn, a player must take one of the three following actions (Give a
  Clue, Discard a Card, or Play a Card).
```

- The other teammates in the game are Teammate 0 and Teammate 1. After your turn,
  Teammate 0 will take their turn, and then Teammate 1 will take their turn after
  Teammate 0.
- After Teammate 1 takes their turn, you will take your turn again, and the game
  continues in this way until the game ends.

## Give a Clue:
- Provide information about either all cards of a specific color or all cards of
  a specific rank in one of your teammates hands.
- The clue will identify every card in the teammates hands that matches the given
  clue information (color or rank).
- This action consumes 1 clue token.

## Discard a Card:
- Choose one card from your hand to discard.
- This action will regain 1 clue token (up to a maximum of 8).
- This action will draw a new card from the deck if one is available.

## Play a Card:
- Choose one card from your hand and attempt to play it on the corresponding
  firework stack.
- If the card is the next rank number in sequence (or a rank 1 card for an empty
  stack), the play is successful and the card is added to the stack.
- If the card played is of rank 5, 1 clue token is regained for the team (if not
  already at the maximum).
- If the card does not match the required sequence, it is a misplay: The team
  loses one life token and the card is discarded.
- The game ends immediately with a score of 0 if all three life tokens are lost
  (bad).
- This action will draw a new card from the deck if one is available.

## Card Positions:
- Slots: Your hand comprises cards with slots 0, 1, 2, 3, 4, with slot 4 as the
  leftmost card and slot 0 as the rightmost. The order of all the slots are:
  - slot 4: leftmost, slot 3: second leftmost, slot 2: middle card, slot 1:
    second rightmost, slot 0: rightmost.
- After play/discard: When a player plays or discards a card from slot X in their
  hand, the card is removed from the hand, and all cards in higher-numbered slots
  are shifted down (to the right) one slot.
- New cards: Any drawn card enters into slot 4 (the leftmost position).

## Game End Conditions:
- When the deck is empty, each player gets one final turn, including the player
  that drew the last card.
- The game ends immediately if all three life tokens are lost, and the players
  get a score of 0 (very bad).
- After the final round (once the deck is exhausted and all players have taken
  their last turn), the game ends.
- A perfect game occurs if all five firework stacks are completed up to card rank
  5 (score 25).

## Scoring:
- If all three life tokens are lost, the game ends with a score of 0 (very bad).

- Otherwise, the final score is the sum of the highest played rank card on each
  firework pile.
- The maximum possible score is 25, achieved when every fireworks stack is
  completed.

```
## Possible Actions:
When you take an action, here are all of the possible actions you may take:
- Discard card in slot 0 from your hand
- Discard card in slot 1 from your hand
- Discard card in slot 2 from your hand
- Discard card in slot 3 from your hand
- Discard card in slot 4 from your hand
- Play card in slot 0 from your hand
- Play card in slot 1 from your hand
- Play card in slot 2 from your hand
- Play card in slot 3 from your hand
- Play card in slot 4 from your hand
- Clue Red to Teammate 0
- Clue Yellow to Teammate 0
- Clue Green to Teammate 0
- Clue White to Teammate 0
- Clue Blue to Teammate 0
- Clue Red to Teammate 1
- Clue Yellow to Teammate 1
- Clue Green to Teammate 1
- Clue White to Teammate 1
- Clue Blue to Teammate 1
- Clue Rank 1 to Teammate 0
- Clue Rank 2 to Teammate 0
- Clue Rank 3 to Teammate 0
- Clue Rank 4 to Teammate 0
- Clue Rank 5 to Teammate 0
- Clue Rank 1 to Teammate 1
- Clue Rank 2 to Teammate 1
- Clue Rank 3 to Teammate 1
- Clue Rank 4 to Teammate 1
- Clue Rank 5 to Teammate 1
```

### A.10.2. H-GROUP CONVENTIONS PROMPT

For experiments with convention-aware agents—and to mirror the play of our human-proxy agent—we include the Level 1 H-Group conventions[1], a small, community-agreed protocol that teaches players to share just enough information to coordinate safely. These rules tell an agent how to signal "this card is safe to play next" or "don't discard this card," and they precisely define which card in your hand each hint refers to. By using these rules, every agent (LLM or human-proxy) speaks the same simple clue vocabulary, letting us measure exactly how much these rules improve teamwork.

```
When you choose an action to take, always use the following convention rules as
the shared protocol for giving and interpreting clues---so you and your teammate
can coordinate discards, saves, and plays unambiguously. The following rules are
written with respect to you, but the same rules apply to your teammates. The
rules are written in the first person, but they apply to all players.
```

[1] https://hanabi.github.io/

# Conventions Rules

## Chop:
- Definition: Your chop is the rightmost card (smallest slot value) in your hand
  that has not received any clues.
- Instructions: When forced to discard, always discard your chop. If a card in
  your hand has been clueed as useless (and you can clearly tell it won't help
  because it's already been played), you may discard that card instead of the
  chop.
- Reminder: If the rightmost slots of your hand (0, 1, etc) have received clues,
  then those slots are NOT your chop, your shop is the rightmost card that has
  not received any clues.

## Clue Interpretation:
- (Card slot position reminder): Your hand comprises cards with 5 different
  slots, with slot 4 as the leftmost card and slot 0 as the rightmost. The order
  of all the slots are:
  - slot 4: leftmost, slot 3: second leftmost, slot 2: middle card, slot 1:
    second rightmost, slot 0: rightmost.
- Single Card Focus: When a clue touches two or more cards, it conveys
  information about only one specific card|the focused card; non-focused cards
  receive no actionable instruction. The focus card must always be either a Save
  Clue or a Play Clue.
- New Cards: Cards are \new" if they had no clues on them prior to this clue.
- Instructions for determining which card is the focus when a clue touches
  multiple cards:
  1. One New Card: If exactly one card is newly clued (had no prior clues), that
     card is the focus of the clue.
  2. Multiple New Cards (chop focus): If more than one card is newly clued and
     the chop is included in the clue, the chop card is the focus.
  3. Multiple New Cards (not including chop): If more than one card is newly
     clued and the chop is not included in the clue, the leftmost new card is the
     focus.
  4. No New Cards: If the clue only touches cards that already had clues, the
     leftmost re-clued card is the focus.
- Clue Type:
  - If the chop is included in the clue, the chop card is the focus and it is
    either a playable card (if it's a Play Clue) or a critical card (if it's a
    Save Clue).
  - If the chop card is not touched by the clue, the focus card (leftmost newly
    clued card) is a Play Clue or a Delayed Play Clue.

## Play Clues:
- Play Clue (Direct):
  - Definition: A clue given to signal that the focused card is immediately
    playable right now (it is the next needed card on its firework stack).
  - Instructions: If the chop card is not touched by a clue, then the focus card
    is always the leftmost card, and it is always a Play Clue. When giving a Play
    Clue, ensure that the focused card will fit directly onto its firework. The
    Play Clue tells your teammate that their focused card is playable right now.
  - Instructions: If the chop card is touched by a clue, then the focus card can
    either be a Play Clue or a Save Clue, it's up to the player to figure out
    which type of clue it is based on what cards are critical.
  - All Play Clues are interpreted as potential Delayed Play Clues.

– Delayed Play Clue:
  – Definition: A clue given to a card that is not immediately playable because
    an earlier card that has received a Play Clue has not been played yet;
    however, once that missing card is played, the clued card will become
    playable.
  – Instructions: When giving a Delayed Play Clue, make sure that the missing
    card will be played soon by either yourself or your teammate. Interpret any
    such clue as a promise that, after the necessary preceding card is played,
    the clued card will be safe to play.

## Critical Cards:
– Definition: A Critical Card is the last copy of a card of a color and rank
  combination that hast not been discarded yet, where discarding this critical
  card makes it impossible to achieve a perfect score.
– Examples: A 5 (only one copy in each color), a unique rank 1 card (if both
  other copies are have been discarded), or any rank 2, 3, or 4 card whose other
  copy has been discarded.
– Instructions: Always treat critical cards as high priority for saving. If a
  critical card becomes the chop card, it must receive a Save Clue to ensure it
  isn't discarded.

## Save Clues:
– Definition: Clues used to protect critical cards from being discarded. Save
  Clues can only be given to cards on the chop.
– Instructions:
  – Save Clues can only be given to cards on the chop.
  – If a clue touches the chp card, it is either a Save Clue or a (Delayed) Play
    Clue. It's up to you to figure out which type of clue it is based on what
    cards are critical.
  – Use Save Clues to safeguard cards that are vital for a perfect score of 25.
  – Critical Save: A clue given to a critical card on the chop to save it from
    being discarded. You can give a Critical Save clue with eithr a color or a
    number clue.
  – 5 Save: When saving a 5 card on the chop, you must always give a rank 5 clue
    to indicate that the clue is a 5 Save, not a color clue because a color clue
    could be interpreted as a play clue.
  – 2 Save: All rank 2 cards on the chop should be saved with a 2 Save clue if
    it's the only copy of that card visible in any players hand, even if 2's are
    not critical. When saving a rank 2 card on the chop, always give a rank 2
    clue to indicate that the card is a 2 Save, a because a color clue could be
    interpreted as a Play Clue.
  – If a clue touches any card that is not on the chop, interpret it as a
    (Delayed) Play Clue, not a Save Clue.
  – When receiving a Save Clue on your chop, do not discard that card|keep it
    safe until it becomes playable.
  – All Save Clues, Critical Save Clues, 5 Saves, and 2 Saves can only be given
    to cards on the chop.

## The Three Main Principles:
1. Good Touch Principle: Only give clues to cards that are have not been played
   yet; avoid clueing cards that have already been played.

2. Save Principle: Including cards that are saved with Save Clues, do not allow other players to discard playable cards. All cards that are playable need to be "protected" by giving them a Play Clue. The following cards must not be discarded: All rank 5 cards, Unique rank 2 cards with only one copy visible, Critical cards with only one copy left undiscarded, and Unique playable cards that are the only copy currently visible.
3. Minimum Clue Value Principle: Every clue must either make one or more cards safely playable or prevent the discard of a critical card. If a clue does not make a card playable or prevent the discard of a critical card, you should discard instead of wasting a clue.

## The Early Game:
- Definition: The phase before any player has ever discarded their chop card.
- Instructions:
  - During this phase, use every available Play Clues and Save Clues to protect critical cards on chop (including 5 and 2 Saves).
  - Only discard if there are no valid Play or Save Clue available.
  - Discarding for the first time ends the Early Game and begins the Mid-Game.

## General Strategy:
- Check Teammate Chops: Always review the rightmost unclued cards in every player's hand to identify which ones need protection.
- Clue Selection: Prefer giving Play Clues when possible over Save Clue, as they not only protect critical cards by giving the player something else to do (playing the card) but also facilitate immediate or near-future plays. Use Save Clues only when a critical card on the chop is at risk of being discarded (the player has nothing else to do on their turn).
- Clue Type: Prefer to use color clues over rank clues for precise information about the card's identity unless a rnak clue can secure additional plays or prevent a vital discard.
- Safe Discards: Discard only cards that are clearly non-critical. Protect any card that might be needed for a perfect score.
- Identify Meaning of Clues: When it's your turn, and you're trying to interpret clue, always follow the follwing algorithm: Identify which card slot in your hand is the focus of the clue, and then decide on if the clue focus card is a Play Clue or a Save Clue.
- Ever clue you give always needs to be a valid Save clue when the card you want to save is on the chop, or a Play/Delayed Play Clue following the rules above.

## Prompts:
- Definition: A Prompt is a clue on a (currently) unplayable card, but it directs a player to immediately play a clued card (which is the card that can be played before the clued card) that they would not normally play right now because they previously didn't have enough information about the cards identity.
- Instructions:
  - The player receiving the prompt to play the connecting card before the clued card can either be the player receiving the Play Clue, or any other player in the team.
  - If more than one card could be interpreted as the card that is being prompted to play, the player being prompted should play the leftmost eligible clued card.
  - When you give a Prompt, the intended action is for the recipient to play the prompted card as soon as possible.

   – Use Prompts sparingly and only when you are confident that the focused card
     is safe to play without further clues.

## Finesse:
- Definition: A move where a clue to one player implies that another player must
  blind-play the connecting lower card (one rank below) from their Finesse
  Position (leftmost unclued slot) even though it has no clues.
- Finesse Position: The card slot containing a player's leftmost unclued card;
  this position slides right whenever leftmost cards receive clues
- Trigger: You see an unplayable card clued as a Play Clue. If the required
  lower-rank card is not visible anywhere with clues (the clue is a prompt),
  assume the player immediately before the clued card holds it in their Finesse
  Position and must blind-play it.
- Priority: Prompts override Finesses. If you must choose between playing a
  prompted card and blind playing a card in finesse position, play the prompted
  card first.
- Urgency: Blind-play into a Finesse immediately to resynchronise team knowledge;
  delaying risks desynchronised information.
- Instructions (for the finessed player that needs to blind-play their finessed
  card):
  1. Identify your current Finesse Position (leftmost unclued slot).
  2. Blind-play that card at once, assuming it is the connecting card 1 rank
    below the clued card.
- Instructions (for the clue-giver):
  – You can only give a finesse clue to Teammate 1, which would trigger Teammate
    0 to blind-play their card in their Finesse Position before Teamamte 1's
    turn.
  – Ensure the clued higher card is unplayable and that the card 1 rank below
    should exist in Teammate 0's Finesse Position.
  – Give the higher card a clue (usually its color or rank) that unmistakably
    marks it as a Play Clue.

### A.10.3. HANABI STATE PROMPT

The final prompt section serialises the simulator observation into an LLM-readable text block: game state counters, public firework stacks, the discard pile, teammates' hands, and information about your own hand. At the end of the prompt we require the LLM to provide its response using a simple JSON response schema. We ask it to fill four fields—game_state, action_options, best_action, and rationale—so that we can understand why it chose the action.

```
# Hanabi Game State

## General Info:
- Game Turn: 39
- Score: 16
- Life Tokens: 3
- Clue Tokens: 0
- Deck Size: 12

## Fireworks:
*(Last card played on each stack, 0 means no card has been played on that stack
  yet)*
- Current Stacks: Red: 3, Yellow: 2, Green: 2, White: 5, Blue: 4

## Discard Pile:
- cards: Red 1, Red 4, Yellow 2, Yellow 3, Yellow 4, Green 4, Blue 1
```

## Teammate 0's Last Action Two Turns Ago:
- Action Type: Clue Rank, Clue: 2, Clue Recipient: Teammate 1, Affected card
  slots in teammate 1's hand: [4]

## Teammate 1's Last Action On The Prevous Turn:
- Action Type: Play, Slot: 4, Card: Yellow 2, Successful Play: Yes

## Your Hand:
*(Card identities unknown to you)*
- Slot 0: Clues(Color: None, Rank: 4), Possible Colors: [Red, Yellow, Green,
  White], Possible Ranks: [4]
- Slot 1: Clues(Color: None, Rank: 5), Possible Colors: [Red, Yellow, Green],
  Possible Ranks: [5]
- Slot 2: Clues(Color: None, Rank: None), Possible Colors: [Red, Yellow, Green,
  White], Possible Ranks: [1, 2, 3, 4, 5]
- Slot 3: Clues(Color: None, Rank: None), Possible Colors: [Red, Yellow, Green,
  White], Possible Ranks: [1, 2, 3, 4, 5]
- Slot 4: Clues(Color: None, Rank: None), Possible Colors: [Red, Yellow, Green,
  White, Blue], Possible Ranks: [1, 2, 3, 4, 5]

## Teammate 0's Hand:
*(Card identities known to you)*
- Slot 0: Card: *Blue 3*, Clues(Color: None, Rank: None), Possible Colors: [Red,
  Yellow, Green, Blue], Possible Ranks: [1, 3, 4]
- Slot 1: Card: *Yellow 4*, Clues(Color: None, Rank: None), Possible Colors:
  [Red, Yellow, Green, Blue], Possible Ranks: [1, 3, 4]
- Slot 2: Card: *Red 1*, Clues(Color: None, Rank: None), Possible Colors: [Red,
  Yellow, Green, White, Blue], Possible Ranks: [1, 2, 3, 4, 5]
- Slot 3: Card: *Green 5*, Clues(Color: None, Rank: None), Possible Colors: [Red,
  Yellow, Green, White, Blue], Possible Ranks: [1, 2, 3, 4, 5]
- Slot 4: Card: *White 4*, Clues(Color: None, Rank: None), Possible Colors: [Red,
  Yellow, Green, White, Blue], Possible Ranks: [1, 2, 3, 4, 5]

## Teammate 1's Hand:
*(Card identities known to you)*
- Slot 0: Card: *Red 4*, Clues(Color: Red, Rank: None), Possible Colors: [Red],
  Possible Ranks: [1, 4, 5]
- Slot 1: Card: *Yellow 3*, Clues(Color: None, Rank: 3), Possible Colors:
  [Yellow, Green, White], Possible Ranks: [3]
- Slot 2: Card: *Blue 5*, Clues(Color: Blue, Rank: None), Possible Colors:
  [Blue], Possible Ranks: [1, 4, 5]
- Slot 3: Card: *White 3*, Clues(Color: None, Rank: None), Possible Colors: [Red,
  Yellow, Green, White, Blue], Possible Ranks: [1, 3, 4, 5]
- Slot 4: Card: *Yellow 1*, Clues(Color: None, Rank: None), Possible Colors:
  [Red, Yellow, Green, White, Blue], Possible Ranks: [1, 2, 3, 4, 5]

## Valid Actions:
*(Choose one action exactly as written)*
- Discard card in slot 0 from your hand
- Discard card in slot 1 from your hand
- Discard card in slot 2 from your hand
- Discard card in slot 3 from your hand
- Discard card in slot 4 from your hand

– Play card in slot 0 from your hand
– Play card in slot 1 from your hand
– Play card in slot 2 from your hand
– Play card in slot 3 from your hand
– Play card in slot 4 from your hand

# Your Instruction:
Please consider all of the Rules and the current Game State, and decide on the best action to take from the "Valid Actions" list.
You cannot play or discard cards from your teammate's hand. Only play a card from your hand if you are certain it is playable based on hints or deduction based on the rules. Check fireworks status before playing. When hints are available for your cards (e.g., 'Colour Hint: Blue' or 'Number Hint: 3'), use them. Evaluate possibilities like 'Colours: [Blue]' or 'Ranks: [3]'. When only one life remains, be extremely cautious about playing cards.

# Response Format
You MUST output responses strictly in JSON format. Adhere EXACTLY to the JSON schema and content requirements provided below.

## JSON Schema:
```json
{
  "game_state": "<str>",
  "action_options": "<str>",
  "best_action": "<str>",
  "rationale": "<str>"
}
```

## Content Requirements:
– "game_state": Provide a summary of the current game state. Include teammate actions, your previous actions, and incorporate any outstanding key reminders from earlier actions, if available, and any important things to taken into account when choosing an action to take. How do you interpret the meaning of the other players previous actions?
– "action_options": List the most promising action options from the Valid Actions list, along with their pros and cons, and why each of them are promising actions. Say the actions exactly as they are written in the Valid Actions list.
– "best_action": Select the best action to play right now. Say the action exactly as it is written in the Valid Actions list.
– "rationale": Provide an explanation for why the selected action is the best choice compared to the other options.

## Instructions:
1. Generate ONLY a single JSON object matching the schema.
2. Do NOT include any text outside the JSON object (e.g., explanations, markdown formatting like ```json, apologies, or status messages).
3. Ensure all required fields from the schema are present in the JSON.

