# OpenReview forum: "Ad-Hoc Human-AI Coordination Challenge"
_ICML.cc/2025/Conference — ICML 2025 spotlightposter_

### Official Review · Reviewer_QVEs · 2025-02-17

**Overall Recommendation:** 2

**Summary:**

The paper introduces the AH2AC2 to evaluate human-AI teamwork in Hanabi, a cooperative card game. Key contributions include:
- AH2AC2 Framework: A standardized benchmark using human proxy agents (trained via behavioral cloning + RL) as evaluation partners, hosted via a controlled API.
- Open-Source Dataset
- Human Proxy Agents: Combining BC with KL-regularized RL, achieving robust performance while retaining human-like behavior.
- Baselines: Evaluation of zero-shot (e.g., OBL) and data-driven methods (e.g., BR-BC), revealing gaps in integrating limited human data.

Main results show OBL outperforms human-data-dependent methods, highlighting challenges in leveraging small datasets. Proxies exhibit human-like behavior via metrics (e.g., IPP, Communicativeness) but lack direct human validation.

**Claims And Evidence:**

Claims are generally supported, however there are some concerns:
- Proxy human-likeness is validated via cross-play with BC, action prediction, and behavioral metrics. However, direct comparison with real human players is absent, weakening claims about human compatibility. Also, overtime, the human behavior may change while the proxy agents will be fixed.
- Superiority of regularized RL over BC is evidenced by improved self-play scores (Table 2).
- Challenge utility is demonstrated via baseline evaluations (Table 5), though reliance on synthetic proxies (not humans) limits real-world applicability claims.

**Essential References Not Discussed:**

No critical omissions noted.

**Experimental Designs Or Analyses:**

- Self/cross-play and action prediction tests are sound but rely on proxy-human comparisons which could be limited in terms of generalization similar to human players. It would be worthwhile to evaluate with human players as well and compare the results.
- Behavioral metrics (IPP, Communicativeness) are validated against a human dataset (Table 4), but metrics are narrow. The paper validates human proxy agents using IPP and Communicativeness, which measure strategic aspects of gameplay (e.g., hint frequency and card knowledge). However, these metrics narrowly focus on specific behaviors, omitting critical dimensions of human-like coordination. For instance, they don't assess risk management (e.g., balancing safe plays versus strategic gambles), temporal adaptation (adjusting strategies as the game progresses), or error recovery (recovering from misplays through ad-hoc coordination)—key facets of human adaptability. Additionally, while prior Hanabi research incorporates broader metrics like hint utility (hint effectiveness) or convention compliance (adherence to implicit rules), the current analysis lacks these, limiting claims of comprehensive human-likeness. Furthermore, the reliance on quantitative metrics overlooks qualitative aspects such as trust or intent alignment, which could be captured through human evaluations (e.g., surveys or A/B testing). Expanding the evaluation to include these dimensions would strengthen assertions about behavioral fidelity and better reflect the complexity of human teamwork.
- Ablation study (Appendix A.8) on regularization strength is critical but not detailed in the main text. I believe it's worth discussing this in more detail in the main paper.

**Methods And Evaluation Criteria:**

- Hanabi is appropriate for testing coordination under partial observability.
- KL-regularized RL is a sensible approach to balance human imitation and robustness.
- API-hosted proxies prevent overfitting, but the limited open dataset (1k games) may restrict research scope, thus the same concern as closed existing evaluation methods would apply to the proposed benchmark.
- Three-player evaluation is underexplained given data scarcity (46k games vs. 101k for two-player).

**Other Comments Or Suggestions:**

In Section 2, discount factor is said to be in range $[0,1]$. How extreme different values affect the results, eg $\gamma = 0$ vs $\gamma =1$?
Minor: in line 163, authors should use `citet`. Same should be applied in several places in the Appendix.

**Other Strengths And Weaknesses:**

- Strengths: Practical framework, reproducible evaluation, actionable insights (e.g., data-efficient methods needed).
- Weaknesses: Limited human validation, sparse three-player data, incremental methodology (BC+RL), limited to only one environment.

**Questions For Authors:**

- Human Validation: How do proxies perform when paired with real humans? If untested, how confident are you in their human-likeness? And how can this be evaluated?
- Three-Player Data: Why prioritize three-player evaluation given limited data? Does data scarcity bias proxy behavior?
- LLM Integration: The conclusion mentions LLMs. What are the plans to benchmark LLM-based agents in AH2AC2?
- Since this is a benchmark paper, it's of utmost importance to keep it updated and maintaining the projects. What are the promises and plans by the authors to do so?

**Relation To Broader Scientific Literature:**

Connects well to Hanabi research (e.g., SPARTA), zero-shot coordination (OBL), and ad-hoc teamwork. Missing recent LLM-based coordination studies (e.g., theory of mind in LLMs), though mentioned in future work.

**Theoretical Claims:**

No new theoretical claims. KL regularization builds on prior work. Theoretical analysis of HDR-IPPO is deferred to future work.

---

> ### Author Rebuttal · Authors · 2025-04-01
>
> Dear reviewer,
>
> Thank you for the detailed and constructive feedback. We appreciate the opportunity to address your points and clarify aspects of our work. Please find our answers and improvements below.
>
> ## Addressing Reviewer Questions
>
> ### Q1. Human Validation
>
> We agree that testing human proxies with real humans would be ideal for a human-AI benchmark, but large-scale evaluations are impractical due to logistical challenges and resource constraints. Therefore, we created these proxies using a method studied extensively in prior research, and our experiments and tests confirm they exhibit **human-like qualities** and behave as expected. Overall, our validation shows that the proxies capture key aspects of human play, making them suitable and robust partners for the benchmark.
>
> ### Q2. Three-Player Data
>
> We thank the reviewer for this excellent question! We collected data for two-, three-, four-, and five-player games; our analysis showed that proxies for four- and five-player settings gave less convincing results due to data sparsity. In contrast, the **three-player setting, with 46k games**, showed strong performance (comparable to the two-player setting), and tests on data subsets revealed diminishing returns with more data. This evidence suggests that the 46k games available for the three-player setting are sufficient to train robust, human-representative proxies, making it a valuable and worthwhile addition to the benchmark.
>
> ### Q3. LLM Integration
>
> Thank you for your suggestion. To strengthen our benchmark and given the popularity of LLMs, we have implemented an LLM-based agent using **o3-mini** and are currently benchmarking it with **AH2AC2**. So far, we're getting scores ranging from **14 to 20 out of 25** (along with some games where all lives are lost), and we'll include these results in the final paper copy, along with additional experiments and results. We believe this is a strong addition to our paper and hope the reviewer will recognise the effort and resources we invested into benchmarking LLMs.
>
> ### Q4. Benchmark Maintenance
>
> We are committed to the long-term success of the AH2AC2 benchmark. We will maintain and update the evaluation website, submission API, and leaderboard, offer ongoing support, and expand the benchmark, potentially adding four and five-player settings and other Hanabi variants as we collect more data and different techniques emerge.
>
> ## Addressing Other Comments
>
> **Fixed proxies and dynamic behavior:**
> Thank you for raising this point. Although our agents’ parameters are fixed after training, their in-game behavior is dynamic – _fixed parameters do not imply fixed behaviour_. Due to space constraints, please see our response to reviewers CEGr and FDiM. Action prediction results (Tables 3 & 4) confirm high accuracy/low loss on unseen human data, indicating they capture diverse strategies and adapt in-game.
>
> **Challenge utility:**
> While we agree that human-AI play is ideal and we acknowledge this limitation, large-scale human testing is often impractical and hard to reproduce. Our approach offers a practical alternative: we create robust, reproducible proxy agents trained on extensive real human gameplay data using _SOTA methods_ for developing human-like policies.
>
> **API-hosted proxies:**
> Thank you for this comment. We believe AH2AC2 improves on previous methods that used completely inaccessible proxies and closed datasets. Our API provides _open_, but _controlled, pre-registered access_ to human proxies, ensuring fair, reproducible evaluation.
>
> **Behavioral metrics:**
> Thank you for your suggestions regarding the breadth of our behavioural evaluation. Our experiments show that BC agents trained on the entire dataset achieve better coordination when paired with our human proxies (*Figure 2*), proving that our proxies adapt and recover from errors even with fixed BC agents. Low **IPP** means hints are effective (they carry implicit info that players must interpret), and our action prediction results (*Tables 3 & 4*) confirm the proxies learn a wide range of implicit conventions.
>
> We couldn’t find standardised definitions for the extra metrics like *hint utility* or *convention compliance* in prior Hanabi works, so we'd be grateful if you could point us to any. We believe our current analysis provides substantial evidence of human-like behavior, and we're open to expanding our evaluation with suitable metrics.
>
> **Ablation study:**
> We appreciate the time you invested in reading the appendix. We completely agree that it should be present in the main text. We will integrate key findings into the main text in the camera-ready version, using an extra page allowance.
>
> ---
>
> We hope these clarifications and planned revisions address your concerns and strengthen our paper. Thank you for your valuable feedback. if you feel our responses have sufficiently resolved your concerns, we kindly ask you to consider updating your score accordingly.

---

### Official Review · Reviewer_FDiM · 2025-03-14

**Overall Recommendation:** 4

**Summary:**

This paper trains a human proxy model from human gameplay records on the Hanabi game and proposes that the proxy model can be used as a cheaper evaluation for algorithms developed for human-AI coordination. They also open-sourced a smaller human dataset on Hanabi.

**Claims And Evidence:**

Yes.

**Essential References Not Discussed:**

No.

**Experimental Designs Or Analyses:**

Yes, I reviewed the experiments in Sec 5.

**Methods And Evaluation Criteria:**

Yes.

**Other Comments Or Suggestions:**

No.

**Other Strengths And Weaknesses:**

Strengths: This paper provides crucial components for human-AI coordination evaluation. The analysis of the human proxy model is comprehensive. The benchmark is a good evaluation for previous human-AI coordination algorithms.
Weaknesses: Since the main purpose is to evaluate human-AI coordination, the most important aspect should be whether the model can reflect real human evaluation results. However, this paper dose not have human study results to show whether the performance against real humans aligns with the performance against human proxy models.

**Questions For Authors:**

1. Humans are known to be diverse. How do you model different strategies with only two proxy models? It is mentioned in line 812 that the dataset  ``adhere to H-Group Conventions’’. Does that constrain the strategy coverage of the dataset?
2. Compared to neural policies, humans are usually much more adaptive. Does that apply to the human dataset in Hanabi? If so, is adaptation captured in the human proxy model?
3. The numbers in table 4 are close, but (since it is not ``normalized’’) it is hard for me to directly see why it shows the behaviors are close.

**Relation To Broader Scientific Literature:**

In human-AI coordination, a good human proxy model is important to evaluate the performance of trained agents. This work provides human proxy models trained from a large human dataset, open-sourced a human dataset, and built a public evaluation platform, which are helpful for future studies on human-AI coordination problems.

**Theoretical Claims:**

This work does not include proofs.

---

> ### Author Rebuttal · Authors · 2025-04-01
>
> Dear reviewer,
>
> Thank you for your positive feedback and detailed review of our paper. We especially appreciate you taking the time to read the appendices. We hope the following answers clarify the points you raised.
>
> ### Question 1
>
> We thank the reviewer for raising this point, and we agree with the point that human play is diverse. Regarding the **H-Group Conventions** mentioned (line 812), we want to clarify what this means in practice and how it relates to strategic variety.
>
> While our dataset mainly features games where **H-Group Conventions** are used, this term doesn't refer to a single strategy. Instead, it is helpful to think of **H-Group Conventions** as a collection of different strategies and techniques that players learn and combine. In fact, conventions are sorted into levels. Each level uses a different set of strategies that are used. Mastering even beginner H-Group levels requires learning several of these. Players often mix and adapt these strategies within a single game – depending on the player's strength. Therefore, the dataset itself naturally contains a variety of playstyles. Our action prediction results (Table 3 & Table 4) show that the trained proxies achieve good accuracy and low cross-entropy loss on unseen human data. This suggests that they have successfully learned to represent multiple strategies and patterns present in the human dataset.
>
> We hope this explanation clarifies that the use of **H-Group Conventions** as a foundation does not overly constrain the strategic variety present in our data, and consequently, in what our proxy models represent. Finally, as a similar concern was raised by multiple reviewers, we will clarify these points in our camera-ready submission.
>
> ### Question 2
>
> Adaptation is a crucial aspect to consider when developing human proxies, and we thank the reviewer for raising this point. To clarify how our proxies address adaptation, we believe it is helpful to consider two adaptation types:
>
> *   **Adaptation within a game:** This refers to adapting actions based on the current game state, partner's moves, and available information within a single game. Our human proxies exhibit this type of adaptation. While the parameters of the neural network policy are fixed after training, behaviour adapts dynamically, much like a human player reacts to changing circumstances. When the proxy agents see an unexpected action or observation, then from that action-observation history onward, they will account for the fact that the other agent uses a different convention. Thus a fixed policy does not imply fixed behaviour.
> *   **Adaptation between games/learning new strategies over time:** This means changing fundamental strategies or learning entirely new conventions. Our proxy models are not designed to do this. We want them to consistently represent the playstyle of the population found in our dataset, providing reproducible and consistent partners for evaluation.
>
> Furthermore, our experiments provide evidence for the proxy's within-game adaptability. As shown in Figure 2, BC agents (which can be quite rigid and brittle) perform better when paired with our human proxies compared to SP. This suggests that our proxies are flexible enough within a game to coordinate effectively even with simpler partners and can adjust to mistakes made by these less sophisticated policies.
>
> We hope this clarifies the specific type of adaptation our human proxy models capture and why they are designed this way for consistent evaluation.
>
> ### Question 3
>
> We are happy to emphasize why we choose our metrics and how we intend them to be interpreted.
> - **IPP (Information per Play)** is normalised to a scale of 0 to 1. A value of 0.44 means players, on average, know slightly less than one “information” (color or rank) about the cards they play.
> - **Communicativeness** is defined as a percentage of turns where a player gives a hint, when it's possible to give a hint.
>
> The fact that the values for these behavioural metrics are close between different models suggests that they exhibit similar tendencies regarding information usage, hint efficiency, and communication frequency. We aimed for these metrics to provide a clear comparison.
>
> Perhaps we misunderstood your concern regarding normalisation? If you could elaborate on what aspect feels unnormalized or difficult to interpret, we would be happy to provide further clarification.
>
> ---
>
> We hope these explanations address your questions. Thank you again for your valuable insights. If our responses have resolved your concerns, we would appreciate it if you would consider increasing the support for our work.

---

### Official Review · Reviewer_CEGr · 2025-03-17

**Overall Recommendation:** 4

**Summary:**

This paper proposes a new ad-hoc human AI co-ordination challenge using the game of Hanabi. The authors have trained a human proxy agent and have created a controlled benchmark environment for researchers to test their new ad-hoc coordination algorithms.

**Claims And Evidence:**

This is a benchmark paper. The main claim is that the human proxy agent approximates human game play. Authors have done several analysis to justify this claim.

**Essential References Not Discussed:**

* It is worth benchmarking multi-task learning agent from Nekoei et al, ICML 2021. (https://arxiv.org/abs/2103.03216)
* It is worth noting that the final future work mentioned in the paper has already been  explored in this recent ICLR paper: https://openreview.net/forum?id=pCj2sLNoJq (of course this was after the ICML submission deadline)

**Experimental Designs Or Analyses:**

Yes. No complaints.

**Methods And Evaluation Criteria:**

Check Q2 for my concern about the proposed evaluation criteria to measure the progress in Human - AI coordination.

**Other Comments Or Suggestions:**

* In page 1, what is "broad-based progress"?

* Setting up API access for evaluation is a great idea to avoid overfitting!

**Other Strengths And Weaknesses:**

Strength:

* Great first step towards designing algorithms for better human-AI coordination.

Weakness:

* Having only one human proxy agent is a weakness.

**Questions For Authors:**

Q1. The main premise of this paper is the collection of a large data set of two-player and three-player games. The authors have decided not to release the dataset and only release a small subset for the researchers to finetune with. However, it is not clear what will stop other researchers from scraping the dataset themselves or just collecting more episodes of human play and training an agent using that. Can you clarify whether this is allowed to participate in the leaderboard?

Q2. The fact that there is only one human proxy agent is very limiting. In practice, the agent might have to deal with multiple humans and each human will have their own style and strategy. Why is it that making progress in this single proxy benchmark would lead to better human AI coordination?

Q3. Why dont you have access to OBL weights for 3-player games? Did you try contacting the authors?

Q4. Authors are testing OBL, OP, and FCP. Another relevant agent that can do better zero-shot coordination that is missing here is the multi-task learning agent from Nekoei et al, ICML 2021. It can also be considered as population based and I would like to see this agent benchmarked here as well.

Q5. Is it right that you train 3 seeds for BC, BR-BC, and HDR-IPPO and then pick the best seed based on the validation set?

**Relation To Broader Scientific Literature:**

The paper is well-positioned in the literature.

**Theoretical Claims:**

There is no theory or proof in the paper.

---

> ### Author Rebuttal · Authors · 2025-04-01
>
> Dear Reviewer,
>
> Thank you for your thorough review and constructive feedback. We have carefully considered your questions and provided our responses below.
>
> **Q1. Benchmark Fairness**
>
> We thank the reviewer for raising this point and agree that it is theoretically possible for researchers to attempt scraping game data or collecting new human data independently, and believe this is a valid concern.
>
> To clarify, the core goal of AH2AC2 is to measure how well methods perform when trained *only* on a limited dataset we provide, and using scraped data would not be fair or allowed. Having in mind that scraping is theoretically possible, as correctly noted by the reviewer, we rely on community transparency. We will strongly encourage participants to make their training code publicly available and reproducible. Also, submissions on the leaderboard will indicate whether reproducible code has been provided (and link to the codebase will be added in this case). We will update our submission system to account for this! We hope this helps reduce reviewers' concerns. Finally, while possible, scraping game data is non-trivial, and crucially, there is currently no open-source dataset of human play available.
>
> We should also note that past benchmarks (such as Overcooked AI) relied solely on trust. We believe our approach offers a more rigorous and fair evaluation protocol – we aimed to reduce the chance of overfitting, but we can't exclude the possibility of unethical practices completely.
>
> We hope this addresses your concerns regarding data usage and the fairness of the benchmark.
>
> ---
>
> **Q2. Human Proxies and Benchmark Goals**
>
> We thank the reviewer for raising this important point. We will update the paper to clarify both the specifics of our agents and the goals of AH2AC2. In particular:
>
> 1.  **Number of Agents:** We developed *four* human proxy agents (two for the 2-player setting and two for the 3-player setting), not a single one. We will make this clearer in the paper.
> 2.  **Diversity of Playstyles:** You are right that real-world human play varies significantly. Our agents were trained on a large dataset of games acquired from the `hanab.live` community. Players on this platform generally follow H-group conventions, but this does not mean they follow a single strategy. Instead, H-group conventions are a collection of diverse strategies and techniques. Players mix and match these based on their skill and the game context, leading to significant variation in play style within the dataset our agents learned from. We agree this is not clear in our original paper, and we will update the paper to provide more details.
>
> Finally, we would like to clarify our goal for the benchmark. Our approach is pragmatic and mirrors a real-world use case:
> *   There is a given (inherently meaningful) human population that an algorithm is supposed to cooperate with.
> *   We assume it is possible to collect a small dataset from this population (as is often the case).
> *   The task is then to develop a method that can do well with the source population while having access to only this small dataset.
>
> Success on our benchmark indicates an algorithm's ability to effectively learn cooperative strategies from limited, representative human data, which we believe is a key step towards building human-compatible agents/systems.
>
> We hope this explanation clarifies the nature of our human proxy agents and the practical relevance of our benchmark.
>
> ---
>
> **Q3. 3P OBL Weights**
>
> We thank the reviewer for reading our paper in detail and raising this point. We contacted several authors of the original OBL paper, and they informed us that they unfortunately no longer have access to the 3P OBL weights and the weights were not released with their original work. Re-implementing OBL in JAX is a complex task, and it is beyond the scope of our current work.
>
> ---
>
> **Q4. Multi-task Learning Agent**
>
> We definitely agree that lifelong learning and multi-task learning algorithms could be highly competitive in AH2AC2. However, given the substantial complexity involved in fully implementing and tuning such an agent, we consider this outside the immediate scope of our benchmark. That said, we encourage and welcome the broader research community to benchmark such methods within AH2AC2, as these promising techniques may significantly enhance human-AI coordination.
>
> ---
>
> **Q5. Seeds**
>
> This is correct.
>
> ---
> Regarding <https://openreview.net/forum?id=pCj2sLNoJq>: We thank the reviewer for highlighting this very relevant ICLR paper. We agree that this recently published method directly addresses one of our proposed future works and represents a valuable addition to our study. We plan to add this approach to our benchmark as soon as possible.
>
> ---
>
> We hope we have addressed the reviewer's comments and concerns. We are happy to discuss any of these points further. If our response has resolved your main concerns, we would be grateful if you would consider updating your score.

---

### Official Review · Reviewer_SVSM · 2025-03-20

**Overall Recommendation:** 2

**Summary:**

The authors present a new test-bench for Human-AI collaborative RL using Hanabi.

**Claims And Evidence:**

The main claim is the creation of a benchmark test which is present but with only one team's submissions. Thus the claims that the benchmark will impact the community or improve RL in general aren't supported.

**Essential References Not Discussed:**

N/A

**Experimental Designs Or Analyses:**

See Methods

**Methods And Evaluation Criteria:**

I'm not convinced the benchmark will work as described. The use of human proxies means that the learning objective is static, Hanabi was originally picked as a target for RL as it requires dynamic planning. Optimal Hanabi agents can update their strategies as they observer the other players. A static (even if complicated) player model doesn't have this property, ditto the human games. How does success on the leader-board show improvements in Human-AI collaboration?

**Other Comments Or Suggestions:**

See above

**Other Strengths And Weaknesses:**

The model training and evaluation looks good, but I think explaining why this paper is relevant to people outside Hanabi researchers would greatly aid the paper. The methods used to train the models do not appear to be novel, and releasing a testing procedure along with new models is also common. So, if the authors can explain the "original and rigorous research of significant interest to the machine learning community" component then this would be good paper, but I feel like I'm missing that.

**Questions For Authors:**

See above

**Relation To Broader Scientific Literature:**

This paper claims to be building support for the literature to grow, but does not display new techniques or methods itself. I'm not sure ICML is the correct venue as a result

**Theoretical Claims:**

N/A

---

> ### Author Rebuttal · Authors · 2025-04-01
>
> Dear Reviewer,
>
> Thank you for your feedback. We appreciate the opportunity to address your concerns and clarify the contributions of our work.
>
> ## Adaptivity of Human-Proxy Agents
>
> You raised an important concern about whether using human proxies, trained using fixed parameters, sufficiently captures the dynamism necessary for Hanabi, where players adjust strategies based on their partners’ behavior. Critically, fixed neural parameters do not imply static behavior; our proxies condition on the history of the game, which includes partner actions. Specifically, when the proxy agents see an unexpected action or observation, from that action-observation history onward, they will account for the fact that the other agent is using a different convention. The proxies are trained using SOTA methods and on a diverse dataset of human gameplay data.
>
> Some concrete evidence that the proxies are capable of adaptivity includes:
>
> -  **Action Prediction Results (Tables 3 & 4):** The proxies accurately predict human actions on unseen human gameplay data, which comes from players of various skill levels, who follow different levels of H-group conventions. Such accuracy requires the ability to infer from a given action-observation history what sort of conventions the human is following.
> -  **Successful Coordination with Brittle Partners (Figure 2):** The proxies achieve high performance even when paired with behavioral cloning agents, which are known to be prone to suboptimal play. The proxies’ success even in such scenarios provides strong evidence that they make adaptations during gameplay to ensure effective coordination.
>
> We have further clarified this in the paper and hope this addresses how the proxies reflect dynamic adaptivity similar to human play.
>
> ## Real-World Relevance of AH2AC2
>
> Success in human-AI collaboration requires the ability to robustly generalize cooperative behaviors from limited human data. As well, the literature requires a standardized testing protocol to rigorously measure algorithmic progress. AH2AC2 directly addresses both of these fronts: we release an experimental suite consisting of carefully validated human-proxy agents, trained on extensive human gameplay data, which serve as robust and reproducible evaluation partners. As well, we open-source a deliberately limited dataset of human games to encourage research specifically on data-efficient coordination algorithms.
>
> ## Novelty and Suitability for ICML
>
> We appreciate the opportunity to clarify our decision to submit this work to ICML. Benchmark creation is explicitly recognized by ICML as a meaningful and rigorous research contribution. In particular, our benchmark required:
>
> -   **An Extensive Original Effort:** Large-scale human data collection, meticulous data cleaning, validation, and rigorous modeling using SOTA methods.
> -  **Novel Evaluation Platform:** Developing reproducible human proxy models, carefully validating them, and establishing a standardized evaluation protocol for human-AI coordination in partially observable, cooperative environments.
>
> To the best of our knowledge, AH2AC2 is the first benchmark to explicitly assess human-AI coordination in complex, partially observable settings using realistic human-proxy agents.
>
> ## Conclusion
> In light of your review, we have modified the paper to make the broader significance of our work more clear. We believe a benchmark such as AH2AC2 is highly necessary for progress in human-AI collaboration, and is to date decidedly lacking in the literature.
>
> Please let us know how we can further improve the paper. Otherwise, if you believe these clarifications have benefited the work, we would appreciate a corresponding update in the score.
>
> Thank you for your continued engagement with the review process.

---

### Decision · Program_Chairs · 2025-05-01

**Decision:**

Accept (spotlight poster)

**Comment:**

The paper introduces a human-like agent to test human-AI coordination, based in the cooperative game Hanabi. The proposed agent is synthetic rather than based directly on human data. The authors provide analyses to validate it is human like. This is a dataset contribution, which will be accessible to the community.

The reviewers raise several questions, but ultimately, it is clear that the contribution, while possessing some limitations, has the potential to be a significant contribution to the community and the work to construct and test the agent performed is substantial.